# Localization, Convexity, and Star Aggregation

**Suhas Vijaykumar**[*]
Statistics Center and Dept. of Economics
Massachusetts Institute of Technology
Cambridge, MA 02139
suhasv@mit.edu

## Abstract

Offset Rademacher complexities have been shown to provide tight upper bounds for the square loss in a broad class of problems including improper statistical learning and online learning. We show that the offset complexity can be generalized to any loss that satisfies a certain general convexity condition. Further, we show that this condition is closely related to both exponential concavity and self-concordance, unifying apparently disparate results. By a novel geometric argument, many of our bounds translate to improper learning in a non-convex class with Audibert's star algorithm. Thus, the offset complexity provides a versatile analytic tool that covers both convex empirical risk minimization and improper learning under entropy conditions. Applying the method, we recover the optimal rates for proper and improper learning with the $p$-loss for $1 < p < \infty$, and show that improper variants of empirical risk minimization can attain fast rates for logistic regression and other generalized linear models.

## 1 Introduction

A central question in statistical learning theory asks which properties of the empirical risk minimizer can be inferred without regard to the underlying data distribution. In particular, the convergence rate of empirical risk minimization has been shown to depend both on distribution-specific properties, such as Tsybakov's margin condition (Tsybakov, 2004), as well as distribution-independent or "global" properties, such as VC dimension of the class and curvature of the loss (Mendelson, 2002).

The local Rademacher complexities of Bartlett et al. (2005) address this issue by giving bounds that are data dependent and also adapt to the global properties of the learning problem. More recently, the offset Rademacher complexities of Rakhlin and Sridharan (2014) and Liang et al. (2015) have been shown to provide similar bounds for the square loss in several contexts. The offset complexity provides a simplified analysis that transparently relates the geometry of the square loss to statistical properties of the estimator.

We extend offset Rademacher complexities to systematically study how curvature of the loss translates to fast rates of convergence in proper and improper statistical learning. We show many results in the literature can be studied in a common geometric framework: from classical results on uniform convexity of the loss (Bartlett et al., 2006; Mendelson, 2002), to newer results on exponential concavity (van Erven et al., 2015; Mehta, 2017) and self-concordance (Bilodeau et al., 2020; Marteau-Ferey et al., 2019).

Remarkably, our analysis applies simultaneously to convex empirical risk minimization and to improper learning with the star algorithm of Audibert (2008). In each case, excess risk is tightly

---

[*]Supported by the MIT Jerry A. Hausman Graduate Dissertation Fellowship

bounded by the offset Rademacher complexity. Thus, we give a clean and unified treatment of fast rates in proper and improper learning with curved loss. Our contributions may be summarized as follows.

1. We generalize the offset Rademacher complexities of Liang et al. (2015) to show that when the loss is $(\mu, d)$-convex (defined below), excess risk is bounded by a corresponding offset Rademacher process.

2. We show in this general setting that the "star algorithm" of Audibert (2008) satisfies an upper bound matching the convex case, allowing us to simultaneously treat convex risk minimization and improper learning under general entropy conditions.

3. We show that $(\mu, d)$-convexity captures the behavior of exponentially concave, uniformly convex, and self concordant losses, leading to fast upper bounds for learning under the $p$-loss, the relative entropy, and the logistic loss.

4. We observe that generalized linear models (glms) may be expressed as learning in a non-convex class subject to the exponentially concave relative entropy loss, and introduce a general procedure—the *regularized star estimator*—which attains fast rates in arbitrary glms from a mixture of at most two proper estimators.

In the case of $k$-ary logistic regression with bounded features $X \in \mathbb{R}^q$ and each predictor norm bounded by $B$, our upper bounds on the excess risk take the form

$$\frac{kq}{n} \cdot \text{polylog}(n, k, B, 1/\rho) \tag{1}$$

with probability at least $1 - \rho$. Thus, in the i.i.d. setting, fast rates that depend only logarithmically on $B$ are achievable by a simple, improper variant of empirical risk minimization.

## 1.1 Relation to Prior Work

This paper examines the *localization* or *fast-rate* phenomenon, where the excess loss vanishes at a rate exceeding the model's uniform law of large numbers (Koltchinskii and Panchenko, 2000; Tsybakov, 2004).

Recently, Liang et al. (2015) showed that localization for the square loss is captured by the so-called *offset Rademacher complexity*. In addition to being analytically straightforward, the offset complexity appears in a minimax analysis of online regression (Rakhlin and Sridharan, 2014), suggesting a unified treatment of online and statistical learning.

In parallel work, van Erven et al. (2015) have showed that exp-concavity of the loss, often studied in online optimization, implies fast rates in the statistical setting via the probabilistic "central condition" (see also Juditsky et al. (2008); Mehta (2017)).

Our work unifies these results. In particular, we show that the offset complexity upper bound can be generalized to arbitrary losses satisfying a certain curvature condition that includes exp-concave losses. These results considerably extend the scope of offset Rademacher complexities in the statistical setting, as we illustrate; we also believe it can refine the minimax analysis of online regression with exp-concave loss, which we leave to subsequent work.

Another noteworthy aspect of our work is its treatment of *improper learning*—that is, minimizing loss compared to the class $\mathcal{F}$ with an estimator that need not belong to $\mathcal{F}$. Unlike prior work, which typically involved aggregation over a finite subset of $\mathcal{F}$ (Rakhlin et al., 2017; Audibert, 2009), we provide a completely unified analysis that simultaneously leads to optimal rates for convex empirical risk minimization and improper learning under general entropy conditions. In particular, we circumvent the issue that passing to the convex hull of certain classes can "blow up" their complexity and lead to slower rates (Mendelson, 2002).

We apply our analysis to generalized linear models (glms), such as logistic regression, where standard procedures surprisingly do not attain fast rates (Hazan et al., 2014). Inspired by Foster et al. (2018), we note that any glm can be recast as learning in a non-convex class subject to the exp-concave relative entropy. The star algorithm therefore attains fast rates for a wide class of generalized linear models, improving on prior results that exploit strong convexity (see e.g. Rigollet (2012)).

Unlike other recent works on improper logistic regression, we do not prove a polynomial time complexity for our procedure, nor do we study the online setting; these are left as open problems for future work. However, our procedure remains of interest for its simplicity—it outputs a fixed mixture of two predictors as opposed to optimizing at each query—and for its applicability to arbitrary glms (Foster et al., 2018; Mourtada and Gaïffas, 2020; Jézéquel et al., 2020).

## 2 Margin inequalities

We begin our discussion with the following definition which extends the classical *uniform convexity* (see e.g. Chidume (2009)), giving a quantitative bound on the extent to which the plane tangent to $f$ at $(y, f(y))$ lies below the graph of $f$.

**Definition 1.** Let $\mu : \mathbb{R} \to \mathbb{R}$ be increasing and convex with $\mu(0) = 0$, and let $d(x, y)$ be a pseudo-metric on the vector space $B$. A differentiable convex function $f : B \to \mathbb{R}$ with gradient $\nabla f$ is $(\mu, d)$-*convex* if for all $x, y \in B$,

$$f(x) - f(y) - \langle \nabla f|_y, x - y \rangle \geq \mu(d(x, y)) \tag{2}$$

In particular, this condition implies that the lower sets of $f$,

$$f_x = \left\{ z \,\middle|\, f(z) \leq f(x) \right\},$$

are strictly convex, and that for any closed and convex $K$ there is a unique $\hat{x} \in \arg\min_{x \in K} f(x)$. We adopt the notation that for a subset $S \subset B$, $\partial S$ denotes the boundary of $S$, $S^\circ = S \setminus \partial S$ denotes the interior, and $\operatorname{co} S$ denotes the convex hull.

**Remark 2.** At this point, we are intentionally vague about the role of the metric, $d$. We shall subsequently see that the choice of $d$ is crucial for producing good upper bounds. The conventional choice is $d(x, y) = |x - y|$ and $\mu(z) = \alpha z^p$, corresponding to $p$-*uniform convexity (with modulus $\alpha$)*. However, our results on self-concordant and exp-concave losses use non-standard choices of $d$.

By a standard argument, given in Lemma 3 below, the condition (2) on the loss implies that the empirical risk minimizer in a convex class satisfies a *geometric margin inequality*. Intuitively, if another hypothesis performs nearly as well as the empirical minimizer in the sample, it must also be $d$-close to the empirical minimizer.

**Lemma 3.** *Let $K$ be a closed, convex subset of $B$. If $f$ is $(\mu, d)$-convex and $\hat{x}$ minimizes $f$ over $K$ then for all $x \in K$*

$$f(x) - f(\hat{x}) \geq \mu(d(x, \hat{x})). \tag{3}$$

*Proof.* Let $z_\lambda = \lambda x + (1 - \lambda)\hat{x}$ be the line segment interpolating $x$ and $\hat{x}$, and put $\varphi(\lambda) = f(z_\lambda)$. By convexity of $K$, we must have $z_\lambda \in K$, hence $\varphi$ is minimized at $\lambda = 0$ by optimality of $\hat{x}$. Hence, $\varphi(\lambda) - \varphi(0) \geq 0$ for all $\lambda$, and $\nabla \varphi$ must satisfy

$$\nabla \varphi|_{\lambda=0} = \langle \nabla f|_{\hat{x}}, x - \hat{x} \rangle \geq 0.$$

Thus,

$$f(x) - f(\hat{x}) \geq f(x) - f(\hat{x}) - \langle \nabla f|_{\hat{x}}, x - \hat{x} \rangle \geq \mu(d(x, \hat{x})),$$

proving the lemma. $\qquad\square$

### 2.1 The star algorithm

We now introduce our first main result, which extends the above analysis to the problem of learning in a non-convex class. Surprisingly, in the general setting of $(\mu, d)$-convexity, an inequality matching (3) is *always* attained by a simple, two-step variant of empirical risk minimization.

Let us now introduce the procedure. Given $x \in S$, the "star hull" of $S$ with respect to $x$ is defined by

$$\operatorname{star}(x, S) = \{\lambda x + (1 - \lambda)s \,|\, s \in S, \lambda \in [0, 1]\}.$$

The procedure may be summarized as follows.

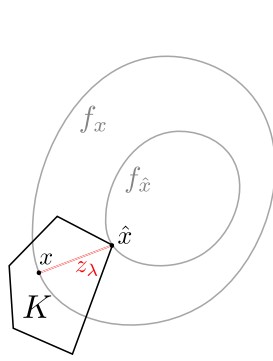

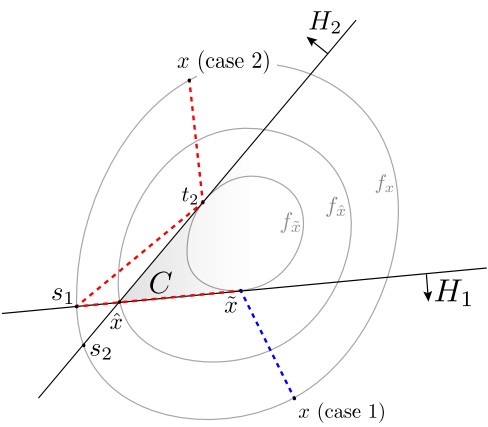

Figure 1: Illustration of Lemma 3. Convexity implies the existence of a direct path $z_\lambda$ from $x$ to $\hat{x}$.

Figure 2: Illustration of Proposition 4. Depicted is the plane containing $(x, \hat{x}, \tilde{x})$; $C$ is the minimal cone containing $f_{\tilde{x}}$ with vertex at $\hat{x}$. By optimality of $\tilde{x}$, $S$ cannot intersect the interior of $C \cup f_{\hat{x}}$, implying existence of one of the indicated paths.

---

1: **procedure** STAR$(S, f)$
2:     Find $\hat{x}$ that minimizes $f$ over $S$
3:     Find $\tilde{x}$ that minimizes $f$ over $\text{star}(\hat{x}, S)$
4:     **Output** $\tilde{x}$
5: **end procedure**

---

Our next goal is to show that if $S$ is an arbitrary (not necessarily convex) set, and $\tilde{x}$ is the output of the star algorithm, then

$$f(x) - f(\tilde{x}) \geq \mu\left(\tfrac{1}{3}d(x, \tilde{x})\right) \tag{4}$$

whenever $f$ is $(\mu, d)$-convex.

In particular, if $f$ is the empirical risk and $S$ is the function class, then $\tilde{x}$ is the "star estimator" introduced by Audibert (2008). Thus, whenever $(\mu, d)$-convexity leads to an upper bound for empirical risk minimization in a convex class via the inequality (3), the same bound is enjoyed up to constants by the star algorithm in an arbitrary class.

To prove (4), recall how we proved Lemma 3: we showed that if the line-segment $\lambda \mapsto z_\lambda$ which linearly interpolates between any two points $x$ and $y$ lies entirely in the set $f_x \setminus f_y^\circ$, so $f(z_\lambda) \geq f(y)$, then we have that

$$f(x) - f(y) \geq \mu(d(x, y)).$$

This is illustrated in Figure 1. Unfortunately, the existence of such a line-segment $z_\lambda$ connecting $x$ and $\tilde{x}$ relies on the convexity of $S$. In order to extend the argument as we want to, we make two observations. Firstly, the line segment $\lambda \mapsto z_\lambda$ can be chosen to be *any* line segment connecting $\partial f_x$ to $\partial f_y$ and lying entirely in $f_x \setminus f_y^\circ$. Then, since $\partial f_x$ and $\partial f_y$ are level sets, the same argument yields

$$f(x) - f(y) = f(z_1) - f(z_0) \geq \mu(d(z_1, z_0)). \tag{5}$$

Secondly, we can show that there exist *at most three* such line-segments which together comprise a path from $x$ to $\tilde{x}$. By the triangle inequality, at least one of these segments has length greater than $\frac{1}{3}d(\tilde{x}, x)$. Combining this with (5) yields the desired inequality (4). We state this as a proposition and prove it formally in Appendix A. A straightforward illustration of the proof, following the preceding discussion, is provided in Figure 2.

**Proposition 4.** *Let $f$ be $(\mu, d)$-convex. Suppose $\hat{x}$ minimizes $f$ over $S$ and $\tilde{x}$ minimizes $f$ over* $\text{star}(\hat{x}, S)$. *Then, for any $x \in S$,*

$$f(x) - f(\tilde{x}) \geq \mu\left(\tfrac{1}{3}d(x, \tilde{x})\right).$$

# 3 From margins to upper bounds

We will now show how the previous section's margin inequalities lead to upper bounds on the excess risk.

## 3.1 Statistical setting and notation

First, let us introduce the statistical setting and some notation. We observe data in the form of $n$ i.i.d. pairs $\{(X_i, Y_i)\}_{i=1}^n \subset D \times \mathbb{R}$ on a common probability space, $\mathbb{P}$. Given a set $\mathcal{F}$ of predictors $f : D \to \mathbb{R}$ and a loss function $\psi : \mathbb{R} \times \mathbb{R} \to \mathbb{R}$, we aim to minimize the excess $\psi$-risk relative to $\mathcal{F}$,

$$\mathcal{E}(\hat{f}; \mathcal{F}, \psi) \doteq \mathbb{E}\psi(\hat{f}(X), Y) - \inf_{f \in \mathcal{F}} \{\mathbb{E}\psi(f(X), Y)]\}.$$

Here, $(X, Y)$ is an independent pair with the same law as the observed data $(X_i, Y_i)$. We will suppress the dependence on $\mathcal{F}$ and $\psi$ where confusion does not arise.

We will use the shorthand $h$ or $h_i$ to denote the random variables $h(X)$ or $h(X_i)$, and will denote the function $(x, y) \mapsto \psi(h(x), y)$ by simply $\psi(h)$. Frequently, we combine these two abuses of notation to write $\psi_i(h) \doteq \psi(h(X_i), Y_i)$. We write $\psi \circ \mathcal{F}$ for the corresponding class of functions $\{\psi(h) \mid h \in \mathcal{F}\}$, and will use $\mathbb{E}_n$ to denote the empirical expectation $\frac{1}{n}\sum_i \delta_{(X_i, Y_i)}$.

## 3.2 Empirical margin inequality

Our first lemma extends $(\mu, d)$-convexity of the loss $x \mapsto \psi(x, y)$ to the empirical risk.

**Lemma 5.** *Suppose that for each $y$, $x \mapsto \psi(x, y)$ is $(\mu, d)$-convex in $x$. Then the empirical $\psi$-risk, given by $f \mapsto \mathbb{E}_n \psi(f)$, is $(\mu, d_n)$-convex in $f$, where $d_n \doteq \mu^{-1} \circ \mathbb{E}_n(\mu \circ d)$.*

*Proof.* By $(\mu, d)$-convexity, we have for each $1 \le i \le n$

$$\psi_i(f) - \psi_i(g) - \langle \nabla\psi|_{g_i}, f_i - g_i \rangle \ge \mu(d(f_i, g_i)).$$

Averaging these inequalities, noting that the inner product and sub-gradient commute with the averaging operation, and also that $\mu \circ \mu^{-1} \circ \mathbb{E}_n(\mu \circ d) = \mathbb{E}_n(\mu \circ d)$, yields Lemma 5. $\qquad\square$

Now, if $\mathcal{F}$ is convex with empirical risk minimizer $\hat{f}$, applying Lemma 3 with $x$ chosen to be the population risk minimizer $f^*$ gives

$$\mathbb{E}_n\psi(f^*) - \mathbb{E}_n\psi(\hat{f}) \ge \mathbb{E}_n\mu(d(\hat{f}, f^*)). \tag{6}$$

Similarly, if $\mathcal{F}$ is not convex and $\tilde{f}$ is produced by the star algorithm applied to $\mathbb{E}_n\psi$, we have by Proposition 4

$$\mathbb{E}_n\psi(f^*) - \mathbb{E}_n\psi(\tilde{f}) \ge \mathbb{E}_n\mu(d(\tilde{f}, f^*)/3). \tag{7}$$

We note that in our applications, $\mu$ will typically have the form $x \mapsto cx^2$, so (6) and (7) are comparable up to a constant factor of 9. To avoid complication, we will only work with the more general inequality (7). The reader may note that in a convex class the star estimator coincides with the empirical risk minimizer, so we are treating both cases simultaneously at the expense of slightly larger constants.

**Example 6** (Square loss). Before proceeding, it is helpful to consider the square loss $\psi(x, a) = (x - a)^2$ as an example. A convenient aspect of $(\mu, d)$-convexity is that the optimal $\mu \circ d$ can be computed as

$$\mu(d(x, y)) = \psi(x) - \psi(y) - \langle \nabla\psi|_y, x - y \rangle$$

$$= \lim_{\lambda \downarrow 0} \frac{\partial}{\partial\lambda} \lambda\psi(x) + (1 - \lambda)\psi(y) - \psi(\lambda x + (1 - \lambda)y),$$

which in this case is simply $(x - y)^2$. Taking $d(x, y) = |x - y|$ and $\mu(z) = z^2$, we recover the familiar inequality

$$\mathbb{E}_n(Y - f^*)^2 - \mathbb{E}_n(Y - \hat{f})^2 \ge \mathbb{E}_n(f^* - \hat{f})^2.$$

Moving to the non-convex case with the star algorithm introduces a factor of $1/9$ on the right-hand side, improving on Liang et al. (2015) by a constant. This analysis extends, with little complication to the $p$-loss for all $1 < p < \infty$. We will discuss these results in Section 5.1.

### 3.3 The offset empirical process

We have now derived a family of margin inequalities for the star estimator,

$$\mathbb{E}_n\psi(f^*) - \mathbb{E}_n\psi(\tilde{f}) - \mathbb{E}_n\mu(d(f^*, \tilde{f})/3) \geq 0. \tag{8}$$

Adding this to the excess risk gives us the upper bound

$$\mathcal{E}(\tilde{f}) = \mathbb{E}\psi(\tilde{f}) - \mathbb{E}\psi(f^*) \leq (\mathbb{E}_n - \mathbb{E})(\psi(f^*) - \psi(\tilde{f})) - \mathbb{E}_n\mu(d(\tilde{f}, f^*)/3)$$

$$\leq \sup_{f \in \mathcal{F}'} \left\{ (\mathbb{E}_n - \mathbb{E})(\psi(f^*) - \psi(f)) - \mathbb{E}_n\mu(d(f, f^*)/3) \right\} \tag{9}$$

for $\mathcal{F}' = \cup_{\lambda \in [0,1]}\lambda\mathcal{F} + (1 - \lambda)\mathcal{F}$ (the Minkowski sum). After symmetrization and contraction arguments, which follow along the lines of Liang et al. (2015), we have the following result. The proof is given in Appendix B.

**Proposition 7.** *Let $\mathcal{F}$ be a model class, $\psi$ a $(\mu, d)$-convex loss, and $f^*$ the population minimizer of the $\psi$-risk. Then, the star estimator $\tilde{f}$ satisfies the excess risk bound*

$$\mathbb{E}\Psi(\mathcal{E}(\tilde{f})) \leq \mathbb{E}\Psi \left( \sup_{f \in \mathcal{F}'} \left\{ \frac{1}{n}\sum_{i=1}^{n} 2\varepsilon_i'(\psi_i(f^*) - \psi_i(f)) - (1 + \varepsilon_i')\mu(\tfrac{1}{3}d(f_i, f_i^*)) \right\} \right) \tag{10}$$

*where the $\varepsilon_i'$ are i.i.d. symmetric Rademacher random variables, $\mathcal{F}' = \cup_{\lambda \in [0,1]}\lambda\mathcal{F} + (1 - \lambda)\mathcal{F}$, and $\Psi : \mathbb{R} \to \mathbb{R}$ is any increasing, convex function.*

**Remark 8.** Proposition 7 quantifies localization for the loss $\psi$. The excess risk is upper bounded by a *non-centered* Rademacher complexity with the asymmetric term $-\mu(d(f_i, f_i^*)/3)$. This term captures the fact that hypotheses $f$ which are far from $f^*$ perform worse in expectation, and are thus unlikely to be selected. As a result, these hypotheses do not contribute much to the excess risk.

**Remark 9.** This bound contains the unknown quantity $f^* \in \mathcal{F}$, and using it requires taking the supremum over possible $f^*$. This leaves open the possibility of exploiting prior information of the form $f^* \in \mathcal{P} \subsetneq \mathcal{F}$. While our bound is stated in expectation, a more elaborate symmetrization argument can provide data-dependent bounds on the excess risk, see Liang et al. (2015).

## 4 Exp-concave localization

We will now investigate how our upper bound (10) applies to $\eta$-exp-concave losses, namely those for which $e^{-\eta\psi}$ is concave, or equivalently

$$\psi''/(\psi')^2 \geq \eta.$$

It turns out that this condition leads to very natural bounds on the excess risk. To provide intuition as to why, note that exploiting the bound (10) requires comparing the risk increments $\psi(f) - \psi(f^*)$ to the offset $\mu(d(f, f^*))$. Under strong convexity, we have an offset of the form $\alpha|f - f^*|^2$ and must resort to a Lipschitz bound of the form $\psi(f) - \psi(f^*) \leq \|\psi\|_{\text{lip}}|f - f^*|$ to make the necessary comparison. Thus, we have upper bounded the first derivative uniformly by $\|\psi\|_{\text{lip}}$ and lower bounded the second derivative uniformly by $\alpha$.

It turns out that exp-concave losses possess an offset term of the form $|\psi(f) - \psi(f^*)|^2$. Thus, *localization naturally occurs at the same scale as the risk increments*. Exp-concavity only requires the ratio $\psi''/(\psi')^2$ to stay bounded, which is crucial for the log loss, where $\|\psi\|_{\text{lip}} = \infty$ but $\psi''/(\psi')^2 = 1$. First, we'll establish a particular version $(\mu, d)$-convexity for the logarithmic loss. We will subsequently extend it to all exp-concave losses.

**Lemma 10.** *For all $x, y \in [\delta, 1]$*

$$(-\ln x) - (-\ln y) - \nabla(-\ln)|_y(x - y) \geq \frac{|\ln x - \ln y|^2}{2\ln(1/\delta) \vee 4}. \tag{11}$$

*Proof.* The left-hand side of (11) is equivalent to $\ln(y/x) + (x - y)/y = e^{-z} - 1 + z$ with the substitution $z \doteq \ln(y) - \ln(x)$. Finally, we verify in Lemma G3 that

$$e^{-z} - 1 + z \geq \frac{z^2}{2\ln(1/\delta) \vee 4}, \tag{12}$$

since our assumption $x, y \in [\delta, 1]$ implies $|z| \leq \ln(1/\delta)$. $\square$

The inequality (11) establishes that the logarithmic loss, when restricted to $[\delta, 1]$, is $(\mu, d)$-convex for $\mu : x \mapsto x^2/(2 \ln(1/\delta) \vee 4)$ and $d(x, y) = |\ln x - \ln y|$. This choice of $d$ captures an important fact: the logarithm is *more convex* precisely when it is *faster varying*.

Remarkably, a simple reduction shows that the same property is enjoyed by bounded, $\eta$-exp-concave losses. Thus, we neatly recover past results on exp-concave learning (cf. Mehta (2017)), and extend them to the improper setting.

**Proposition 11.** *Suppose $\psi$ is an $\eta$-exp-concave loss function which satisfies the uniform bound $0 \leq \psi \leq m$. Then, for each $x$ and $y$,*

$$\psi(x) - \psi(y) - \langle \nabla\psi|_y, \, x - y \rangle \geq \frac{|\psi(x) - \psi(y)|^2}{2m \vee (4/\eta)}. \tag{13}$$

*Proof.* First note that since $e^{-\eta\psi}$ is concave, we have

$$\exp(-\eta\psi(\lambda x + (1 - \lambda)y)) \geq \lambda e^{-\eta\psi(x)} + (1 - \lambda)e^{-\eta\psi(y)}.$$

Note that equality holds at $\lambda = 0$. Taking logarithms of both sides and passing to the upper sub-gradient at $\lambda = 0$ therefore yields

$$-\eta \langle \nabla\psi|_y, \, x - y \rangle \geq \left\langle \nabla \ln|_{e^{-\eta\psi(y)}}, \, e^{-\eta\psi(x)} - e^{-\eta\psi(y)} \right\rangle. \tag{14}$$

We then apply Lemma 10 with $x$ and $y$ replaced by $e^{-\eta\psi(x)}$ and $e^{-\eta\psi(y)}$, and with $\delta = e^{-\eta m}$. Using (14) to bound the gradient term then yields Proposition 11. □

Plugging (13) into Proposition 7 and making a contraction argument yields the following upper bound for the excess $\psi$-risk. The full argument is in Appendix C.

**Theorem 12.** *Let $\psi$ be an $\eta$-exp-concave loss function taking values in $[0, m]$. Then the star estimator in $\mathcal{F}$ satisfies the excess risk bound*

$$\mathbb{E}\Psi(\mathcal{E}(\tilde{f})) \leq \mathbb{E}\Psi\left( \sup_{f,g\in\mathcal{F}'} \left\{ \frac{1}{n} \sum_{i=1}^{n} 4\varepsilon_i'(\psi_i(f) - \psi_i(g)) - \frac{\eta(\psi_i(f) - \psi_i(g))^2}{18m\eta \vee 36} \right\} \right). \tag{15}$$

*where $\Psi$ is any increasing, convex function and $\mathcal{F}' = \cup_{\lambda \in [0,1]} \lambda\mathcal{F} + (1 - \lambda)\mathcal{F}$. Alternatively, when $\psi$ is $p$-uniformly convex with modulus $\alpha$ and $\|\psi\|_{\mathrm{lip}}$-Lipschitz, we have*

$$\mathbb{E}\Psi(\mathcal{E}(\tilde{f})) \leq \mathbb{E}\Psi\left( \sup_{f,g\in\mathcal{F}'} \left\{ \frac{1}{n} \sum_{i=1}^{n} 4\|\psi\|_{\mathrm{lip}}(f_i - g_i)\varepsilon_i' - \frac{\alpha|f_i - g_i|^p}{3^p} \right\} \right). \tag{16}$$

**Remark 13.** An especially nice feature of (15), as well as (18) below, is that all quantities appearing in the bound are invariant to affine re-parameterizations of the learning problem.

**Self-concordance**

We close the section by considering *self-concordant* losses $\psi$, namely those for which

$$\psi''' \leq 2(\psi'')^{\frac{3}{2}}.$$

In order to see how localization for self-concordant losses can be fit into the above framework, we rely on the following inequality which was also used by Bilodeau et al. (2020).

**Lemma 14** (Nesterov (2014, Theorem 4.1.7)). *Suppose $\psi$ is self-concordant, and define the local norm $\|z\|_{\psi,w} \doteq \sqrt{z^2 \psi''(w)}$. Then*

$$\psi(x) - \psi(y) - \langle \nabla f|_y, \, x - y \rangle \geq \|x - y\|_{\psi,y} - \ln\left(1 + \|x - y\|_{\psi,y}\right). \tag{17}$$

The first thing that one should note is that, when one takes $\psi$ to be the negative logarithm, the above is exactly equivalent to the modulus (12) appearing in the proof of Lemma 10. Thus, our upper bound for exp-concave losses is a consequence of self-concordance of the logarithm.

The path to general upper bounds via self-concordance is significantly more involved, so we will only sketch it. Applying (17) to the excess risk and performing standard manipulations, one arrives at the following upper bound, proved in Appendix D.

**Proposition 15.** *If $\psi$ is a self-concordant loss and $\hat{f}$ is the empirical risk minimizer in a convex class $\mathcal{F}$, then*

$$\mathbb{E}\Psi(\mathcal{E}(\tilde{f})) \leq \mathbb{E}\Psi\left(\sup_{f\in\mathcal{F}'}\left\{\frac{1}{n}\sum_{i=1}^{n} 4(\psi_i(f) - \psi(f_i^*))\varepsilon_i' - \omega\left(\|f_i - f_i^*\|_{\psi,f_i^*}\right)\right\}\right), \quad (18)$$

*for $\omega(z) = z - \log(1+z)$, $\|z\|_{\psi,w} \doteq \sqrt{z^2\psi''(w)}$, and $(\varepsilon_i')_{i=1}^n$ are independent, symmetric Rademacher random variables and $\Psi$ is any increasing, convex function.*

Comparing this to (16), the scale of the offset term has been improved from the modulus of strong convexity, which is the *minimum* second derivative, to the second derivative at the risk minimizer, $f^*$. Expressing the increments $\psi_i(f) - \psi_i(f^*)$ at a scale comparable to $\|f_i - f_i^*\|_{\psi,f^*}$ is known to be achievable when $f$ belongs to the so-called *Dikin ellipsoid* of $(\mathbb{E}_n\psi, f^*)$, requiring a more careful argument (see e.g. Marteau-Ferey et al. (2019)). Notably, we do not know how to extend this bound to the improper setting due to the local nature of the offset term.

## 5 Applications

We will now discuss several applications of Theorem 12. First, we'll sketch how the previous section's results straightforwardly imply fast rates. We'll then discuss applications, first to the $p$-loss and then to generalized linear models. Proofs in both cases are to be found in Appendix F.

While most general bound uses chaining, and is provided by Theorem 19 below, the essence of the argument is simple; we will sketch it now. Let $S_\epsilon(\mathcal{A}, \nu)$ be a minimal covering of $\mathcal{A}$ in $L^2(\nu)$ at resolution $\epsilon$, and define the entropy numbers $H_2(\epsilon, \mathcal{G}) \doteq \sup_\nu \ln(\#S_\epsilon(\mathcal{G}, \nu))$ (where the supremum is over probability measures $\nu$).[2] Putting $S_\epsilon \doteq S_\epsilon(\psi \circ \mathcal{F}', \mathbb{P}_n)$ and applying (15) with $\Psi(z) = \exp(\lambda n z)$ gives

$$\mathbb{E}\exp(\lambda n(\mathcal{E}(\tilde{f}) - \epsilon)) \leq \mathbb{E}\exp\lambda n\left(\sup_{s,t\in S_\epsilon}\left\{\frac{1}{n}\sum_{i=1}^{n} 4\varepsilon_i'(s_i - t_i) - \frac{\eta(s_i - t_i)^2}{18m\eta \vee 36}\right\}\right)$$

$$\leq \mathbb{E}\sum_{s,t\in S_\epsilon}\mathbb{E}_\varepsilon \exp\left(\sum_{i=1}^{n} 4\lambda\varepsilon_i'(s_i - t_i) - \frac{\lambda\eta(s_i - t_i)^2}{18m\eta \vee 36}\right),$$

where we first moved the supremum outside $\exp$ (by monotonicity of $\exp$) and then bounded it by a sum (by positivity of the arguments). Here, $\mathbb{E}_\varepsilon$ denotes an expectation w.r.t. the multipliers $\varepsilon_i'$, conditional upon the data. Applying Hoeffding's lemma to the inner expectations with respect to $\varepsilon_i'$ gives

$$\leq \mathbb{E}\sum_{s,t\in S_\epsilon}\exp\left(\sum_{i=1}^{n} 2\lambda^2(s_i - t_i)^2 - \frac{\lambda\eta(s_i - t_i)^2}{18m\eta \vee 36}\right).$$

Choosing $\lambda^* = 36m \vee 72/\eta$ gives $\mathbb{E}\exp(\lambda^*n(\mathcal{E}(\tilde{f}) - \epsilon)) \leq \mathbb{E}(\#S_\epsilon)^2 \leq \exp\{2H_2(\epsilon, \psi \circ \mathcal{F}')\}$, so by a Chernoff bound

$$\mathbb{P}\left(\mathcal{E}(\tilde{f}) \geq \epsilon + \left(36m \vee \frac{72}{\eta}\right)\left(\frac{2H_2(\epsilon, \psi \circ \mathcal{F}') + \ln(1/\rho)}{n}\right)\right) \leq \rho. \quad (19)$$

The result (19) extends Mehta (2017, Theorem 1) to the non-convex and improper setting for the price of a factor 9, and differs from the standard bound in numerous ways. Firstly, the Lipschitz constant $\|\psi\|_{\text{lip}}$ is replaced by the range, $m$. Here, $\|\psi\|_{\text{lip}}$ appears only implicitly in the covering numbers, entering logarithmically in small classes. For the log loss, this is an exponential improvement.

Next, the modulus of strong convexity is replaced by the exp-concavity modulus $\eta$. This implies that the optimal $1/n$ rate can be obtained, in both proper and improper settings, for losses which are not strongly convex but are exp-concave, notably regression with $p$-loss for $p > 2$.

---

[2]Our results can be sharpened by restricting the supremum to empirical probability measures on $n$ data points. With further work, it is also possible to derive high-probability bounds where $\nu$ is restricted to be the empirical distribution of the observed data, see Remark 9.

Lastly, we show in Appendix Lemma G4 that the entropy numbers of $\psi \circ \mathcal{F}'$ are equivalent to those of $\psi \circ \mathcal{F}$ in all but finite classes, where the gap is at most $\ln(1/\epsilon)$. On the other hand, when the entropy numbers of $\psi \circ \mathcal{F}$ are polynomial, particularly $\epsilon^{-q}$ with $q < 2$, then they can be polynomially smaller than those of $\psi \circ (\mathrm{co}\,\mathcal{F})$ (Carl et al., 1999).

## 5.1 Regression with $p$-loss

A surprising consequence of (19) concerns regression with the $p$-loss for $p > 2$, which is $p$-uniformly convex but *not* strongly convex. When the inputs to the loss are bounded, the standard argument pursued by Mendelson (2002) and Bartlett et al. (2006) (and also Rakhlin and Sridharan (2015) in the online setting) exploits uniform convexity, producing the suboptimal rate exponent $p/(2p - 2) < 1$. However, the $p$-loss for inputs in $[-B, B]$ is $B^{-p}$-exp-concave, so we recover optimal $(1/n)$-type rates matching the square loss (cf. Audibert (2009)). Indeed, the suboptimal rates occur exactly when one plugs (16) into the above argument instead of (15). For $1 < p \leq 2$, the loss is both strongly convex and exp-concave, so the two bounds coincide up to constants.

**Corollary 16** (cf. Mendelson (2002, Theorem 5.1)). *Let $\psi(f, y) = |f - y|^p$ for $p > 1$ and let the class $\mathcal{F}$ and response $Y$ take values in $[-B, B]$. Then there exists a universal $C(p, B) = O(p2^p B^p)$ such that the star estimator $\tilde{f}$ has excess $\psi$-risk bounded as*

$$\mathbb{P}\left( \mathcal{E}(\tilde{f}, \mathcal{F}) \geq \epsilon + \frac{C(p, B)(H_2(\epsilon/C_{p,B}, \mathcal{F}) + \ln(1/\epsilon) + \ln(1/\rho))}{n} \right) \leq \rho, \qquad (20)$$

By a minor extension to Tsybakov (2003, Theorem 1), this result is sharp for each $p$.

## 5.2 Improper maximum likelihood

We next derive upper bounds for learning under the log loss, $\psi : f \mapsto -\ln f$ in a class $\mathcal{L}$. If we interpret $f$ as the likelihood of an observation, empirical risk minimization coincides with maximum likelihood.

In this setting, a generalized linear model for multi-class prediction refers to the case where $\mathcal{L} = \{(x, y) \mapsto \langle \varphi(f), y \rangle \mid f \in \mathcal{F}\}$ for some base class $\mathcal{F}$. Here the *link function* $\varphi$ outputs probability distributions over $[k]$, and $y \in \mathbb{R}^k$ is a basis vector encoding the label.

These models are typically formulated as minimizing the loss $\psi^{[\varphi]} : f \mapsto -\ln \langle \varphi(f), y \rangle$ over $f \in \mathcal{F}$, which is equivalent for proper learning. Since $\mathcal{L}$ may be non-convex even for convex $\mathcal{F}$, however, one must consider improper predictors $\hat{f} \notin \mathcal{L}$ in order to exploit localization of the log loss. When $\varphi$ is surjective, these yield improper predictors under $\psi^{[\varphi]}$ by composing with a right-inverse, $\varphi^{\dagger}$.

To illustrate, we will derive our quoted bound (1) for logistic regression. In this setting, Hazan et al. (2014) showed that improper learning is necessary for fast rates with sub-exponential dependence on the predictor norm, $B$. This was achieved by Foster et al. (2018), whose results fall neatly into the aforementioned recipe: they propose a mixed prediction corresponding to an exponentially weighted average, which is simply another estimator that takes values in the convex hull of $\mathcal{L}$.

For classes not bounded away from 0, where the improvement is most dramatic, we need to replace $\mathcal{L}$ by the $\delta$-regularization $\mathcal{L}_\delta = (1 - \delta)\mathcal{L} + \delta$. The approximation error is controlled as follows.

**Lemma 17** (Foster et al. (2018)). *For all $f$ and $\delta \in (0, 1/2]$, the excess risk relative to $\mathcal{L}_\delta$ satisfies*

$$\mathcal{E}(f; \mathcal{L}_\delta) \leq \mathcal{E}(f; \mathcal{L}) + 2\delta$$

We now state our result.

**Corollary 18.** *With probability at least $1 - \rho$, the star estimator $\tilde{f}_\delta$ in $\mathcal{L}_\delta$ satisfies*

$$\mathcal{E}(\tilde{f}_\delta; \mathcal{L}) \leq \epsilon + 2\delta + C\ln(1/\delta)\left( \frac{H_2(\delta\epsilon, \mathcal{L}) + \ln(1/\epsilon\delta) + \ln(1/\rho)}{n} \right) \qquad (21)$$

*Let $\mathcal{L}$ be the generalized linear model corresponding to*

$$\mathcal{F}_B = \left\{ x \mapsto Wx \,\middle|\, W \in \mathbb{R}^{k \times q}, \|W\|_{2 \to \infty} \leq B \right\}$$

with *A-Lipschitz, surjective link $\varphi$ and features $X \in \mathbb{R}^q$ that satisfy $\|X\|_2 \leq R\sqrt{q}$. Then with probability at least $1 - \rho$*

$$\mathcal{E}(\varphi^\dagger \circ \tilde{f}_\delta; \mathcal{F}_B) \leq \frac{\ln(n)}{n} \left\{ Ckq \ln(ABRn\sqrt{k}) + \ln(1/\rho) \right\}. \tag{22}$$

Our bound for logistic regression, quoted in the introduction, follows after noting that the soft-max function is 1-Lipschitz and surjective.

Before investigating how these bounds translate to larger classes—namely those with $H_2(\epsilon, \mathcal{F})$ of order $\epsilon^{-q}$ for small $\epsilon$—we need to state the chaining version of our bound, proved in Appendix E.

**Theorem 19.** *Let $\psi$ be an $\eta$-exp-concave loss taking values in $[0, m]$. Then, with probability at least $1 - 9e^{-z}$, the star estimator $\tilde{f}$ applied to $(\psi, \mathcal{F})$ satisfies*

$$\mathcal{E}(\tilde{f}) \leq \inf_{0 \leq \alpha \leq \gamma} \left\{ 4\alpha + \frac{10}{\sqrt{n}} \int_\alpha^\gamma \sqrt{H_2(s)} \, ds + \frac{2H_2(\gamma)}{cn} + \frac{\gamma\sqrt{8\pi}}{\sqrt{n}} + \left( \frac{2}{cn} + \frac{\gamma\sqrt{8}}{\sqrt{n}} \right) z \right\}, \tag{23}$$

*where $H_2(s) \doteq H_2(s, \psi \circ \mathcal{F}')$ and $c = 36^{-1}(1/m \wedge \eta/2)$.*

In classes with entropy $\epsilon^{-q}$, our results fall into two regimes. The first covers losses such as the logistic loss, for which $-\ln \circ \varphi$ is $A$-Lipschitz. Here, the covering numbers of $-\ln \circ \mathcal{L}_\delta$ differ from those of $\mathcal{F}$ by the constant $A^q$, and we match the optimal rates for the square loss (Rakhlin et al., 2017).

For general $\mathcal{L}$, the covering numbers of $-\ln \circ \mathcal{L}_\delta$ may contain an extra factor $\delta^{-q}$, leading to suboptimal rates. Suboptimality follows from combining the minimax regret analysis of Bilodeau et al. (2020), which produces the average regret $n^{-1/(1+q)}$, with a standard online-to-batch reduction. We note that the upper bound of that paper is non-constructive, whereas these upper rates come with an explicit algorithm. Tightening our analysis to close this gap, or proposing an algorithm that attains the correct rates, remains a significant open problem.

**Corollary 20.** *Consider a generalized linear model with $A$-Lipschitz loss $f \mapsto -\ln \langle \varphi(f), y \rangle$. Suppose the entropy numbers $H_2(\epsilon; \mathcal{F})$ are of order $\epsilon^{-q}$. Then, the regularized star estimator $\varphi^\dagger \circ \tilde{f}_\delta$ with $\delta = 1/n$ satisfies the rates appearing on the left.*

*On the other hand, for an arbitrary class $\mathcal{L}$ taking values in $[0, 1]$ subject to the log loss, the regularized star estimator $\tilde{f}'_{\delta'}$—for appropriately chosen $\delta'$—attains the rates appearing on the right.*

*Here, the symbol $\lesssim_\rho$ denotes an upper bound that holds with probability $1 - \rho$, hiding universal constants and a multiplicative factor $\ln(1/\rho)$.*

$$\mathcal{E}(\varphi^\dagger \circ \tilde{f}_\delta) \lesssim_\rho \begin{cases} A^q n^{-2/(2+q)} \ln(n) & q < 2 \\ A^q n^{-1/2} \ln(n) & q = 2 \\ A^q n^{-1/q} & q > 2 \end{cases} \quad \mathcal{E}(\tilde{f}'_{\delta'}) \lesssim_\rho \begin{cases} n^{-1/(1+3q/2)} & q < 2 \\ n^{-1/4} \ln(n) & q = 2 \\ n^{-1/(2q)} & q > 2 \end{cases} \tag{24}$$

In closing, we would like to highlight the additional, unresolved problem of designing a computationally efficient implementation of the regularized star estimator—in particular for log-concave link functions $\varphi$ where the first step is a convex program, such as logistic regression. This problem is left to future work.

## Acknowledgments and Disclosure of Funding

This work was partially funded by the MIT Jerry A. Hausman Graduate Dissertation Fellowship. We would like to thank Alexander Rakhlin, Victor Chernozhukov, and Anna Mikusheva for their extensive guidance and for many helpful conversations. We would also like to thank Stefan Stein and Claire Lazar Reich for their helpful feedback on the manuscript. All errors are, however, the fault of the author.

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
