# Appendix

## Table of Contents

## A    Proof of Proposition 4

We recall Proposition 4

**Proposition 4.** *Let $f$ be $(\mu, d)$-convex. Suppose $\hat{x}$ minimizes $f$ over $S$ and $\tilde{x}$ minimizes $f$ over* $\mathrm{star}(\hat{x}, S)$. *Then, for any $x \in S$,*

$$f(x) - f(\tilde{x}) \geq \mu \left( \tfrac{1}{3} d(x, \tilde{x}) \right).$$

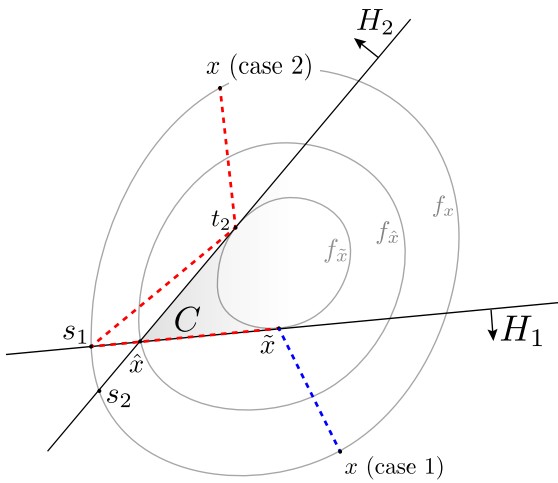

Figure 3: Illustration of Proposition 4.

*Proof.* The proof relies on the observation that if $\lambda \mapsto z_\lambda$ is a line segment such that $z_0 \in \partial f_x$ (i.e. $f(z_0) = f(x)$), $z_1 \in \partial f_y$ (i.e. $f(z_1) = f(y)$), and $z_\lambda \in f_x \setminus f_y^\circ$ (i.e. $f(x) \geq f(z_\lambda) \geq f(y)$) for $\lambda \in [0, 1]$, then

$$f(x) - f(y) = f(z_0) - f(z_1) \geq \mu(d(z_0, z_1)).$$

This is seen by repeating the proof of Lemma 3: since $z_\lambda \notin f_y^\circ$ we must have $(f(z_\lambda) - f(z_1))/\lambda \geq 0$. Plugging in $z_\lambda = z_0 + \lambda(z_1 - z_0)$ and taking the limit infimum as $\lambda \downarrow 0$ gives $0 \leq \langle \nabla f|_{z_1}, z_0 - z_1 \rangle$. By $(\mu, d)$-convexity of $f$, then,

$$f(z_1) - f(z_0) \geq f(z_1) - f(z_0) - \langle \nabla f|_{z_1}, z_0 - z_1 \rangle \geq \mu(d(z_1, z_0)).$$

Finally, if $k$ such segments $z_\lambda$ form a path from $x$ to $y$, then at least one of them must have $d(z_1, z_0) \geq \frac{1}{k}d(x, y)$. This is due to the triangle inequality for $d$.

We now restrict our attention to the plane $P$ containing $(x, \hat{x}, \tilde{x})$. Let $C$ be the minimal cone containing $f_{\tilde{x}}$ with vertex at $\hat{x}$. We note that, by optimality of $\tilde{x}$, no point $x \in S$ can lie in the interior of $C$.

Then $C \cap P$ is complementary to the union of two half-planes $H_1$ and $H_2$ with boundary lines $\ell_1$ and $\ell_2$, respectively. These lines intersect $\partial f_x$ at two respective points, $s_1$ and $s_2$, and are tangent to $f_{\tilde{x}}$ at two respective points, $t_1 \equiv \tilde{x}$ and $t_2$. Finally, $f_x \supseteq f_{\hat{x}}$ by optimality of $\hat{x}$. This is depicted in Figure 3 above.

There are two cases to consider. In the first case, $x \in H_1$. Then the line segment connecting $\tilde{x}$ and $x$ is contained entirely in $D = f_x \setminus f_{\tilde{x}}^\circ$. By the preceding discussion,

$$f(x) - f(\tilde{x}) \geq \mu(d(x, \tilde{x})) \geq \mu\left(\tfrac{1}{3}d(x, \tilde{x})\right).$$

In the second case $x \in H_2 \setminus H_1$. In this case, the segment from $\tilde{x}$ to $s_1$ along $\ell_1$ lies entirely in $D$. Similarly, the segments from $s_1$ to $t_2$ and from $t_2$ to $x$ are line segments contained in $D$. Each of these three segments connects $\partial f_{\tilde{x}}$ to $\partial f_x$, and they together form a path from $\tilde{x}$ to $x$. By the preceding discussion,

$$f(x) - f(\tilde{x}) \geq \mu\left(\tfrac{1}{3}d(x, \tilde{x})\right).$$

This completes the proof. $\qquad\square$

# B   Proof of Proposition 7

We recall Proposition 7:

**Proposition 7.** *Let $\mathcal{F}$ be a model class, $\psi$ a $(\mu, d)$-convex loss, and $f^*$ the population minimizer of the $\psi$-risk. Then, the star estimator $\tilde{f}$ satisfies the excess risk bound*

$$\mathbb{E}\Psi(\mathcal{E}(\tilde{f})) \leq \mathbb{E}\Psi\left(\sup_{f \in \mathcal{F}'}\left\{\frac{1}{n}\sum_{i=1}^{n} 2\varepsilon_i'(\psi_i(f^*) - \psi_i(f)) - (1 + \varepsilon_i')\mu(\tfrac{1}{3}d(f_i, f_i^*))\right\}\right) \qquad (10)$$

*where the $\varepsilon_i'$ are i.i.d. symmetric Rademacher random variables, $\mathcal{F}' = \cup_{\lambda \in [0,1]}\lambda\mathcal{F} + (1 - \lambda)\mathcal{F}$, and $\Psi : \mathbb{R} \to \mathbb{R}$ is any increasing, convex function.*

*Proof.* We'll work forwards from (9), which says that

$$\mathcal{E}(\tilde{f}) = \mathbb{E}\psi(\tilde{f}_i) - \mathbb{E}\psi(f^*)_i \leq \sup_{f \in \mathcal{F}'}\left\{(\mathbb{E}_n - \mathbb{E})(\psi(f_i^*) - \psi(f_i)) - \mathbb{E}_n\mu(d(f_i, f_i^*)/3)\right\}$$

Using the notation $\Delta_i(f) = \psi(f_i^*) - \psi(f)_i$, $\nu_i(f) = \frac{1}{2}\mu(\tfrac{1}{3}d(f_i, f_i^*))$, we can rewrite this as

$$\mathcal{E}(\tilde{f}) \leq \sup_{f \in \mathcal{F}'}\left\{\frac{1}{n}\sum_{i=1}^{n}(1 - \mathbb{E})\Delta_i(f) - 2\nu_i(f)\right\}.$$

Adding and subtracting $\mathbb{E}\nu_i(f)$ gives

$$= \sup_{f \in \mathcal{F}'}\left\{\frac{1}{n}\sum_{i=1}^{n}(1 - \mathbb{E})(\Delta_i(f) - \nu_i(f)) - (1 + \mathbb{E})\nu_i(f)\right\}.$$

By applying $\mathbb{E}\Psi$ to both sides and applying Lemma G1 (a symmetrization result along the lines of Liang et al. (2015)) with $A = \Delta$, $B = \nu$, and $T = \mathcal{F}$, we get

$$\leq \mathbb{E}\Psi\left(2\sup_{f \in \mathcal{F}'}\left\{\frac{1}{n}\sum_{i=1}^{n}\varepsilon_i'(\Delta_i(f) - \nu_i(f)) - \nu_i(f)\right\}\right)$$

$$= \mathbb{E}\Psi\left(\sup_{f \in \mathcal{F}'}\left\{\frac{1}{n}\sum_{i=1}^{n} 2\varepsilon_i'(\psi(f^*)_i - \psi(f)_i - \tfrac{1}{2}\mu(\tfrac{1}{3}d(f_i, f_i^*))) - \mu(\tfrac{1}{3}d(f_i, f_i^*))\right\}\right),$$

where we plugged in the definition of $\Delta_i$ and $\nu_i$. This completes the proof. $\qquad\square$

# C Proof of Theorem 12

We recall Theorem 12:

**Theorem 12.** *Let $\psi$ be an $\eta$-exp-concave loss function taking values in $[0, m]$. Then the star estimator in $\mathcal{F}$ satisfies the excess risk bound*

$$\mathbb{E}\Psi(\mathcal{E}(\tilde{f})) \leq \mathbb{E}\Psi\left(\sup_{f,g\in\mathcal{F}'}\left\{\frac{1}{n}\sum_{i=1}^{n}4\varepsilon_i'(\psi_i(f) - \psi_i(g)) - \frac{\eta(\psi_i(f) - \psi(g_i))^2}{18m\eta \vee 36}\right\}\right). \qquad (15)$$

*where $\Psi$ is any increasing, convex function and $\mathcal{F}' = \cup_{\lambda\in[0,1]}\lambda\mathcal{F} + (1-\lambda)\mathcal{F}$. Alternatively, when $\psi$ is p-uniformly convex with modulus $\alpha$ and $\|\psi\|_{\mathrm{lip}}$-Lipschitz, we have*

$$\mathbb{E}\Psi(\mathcal{E}(\tilde{f})) \leq \mathbb{E}\Psi\left(\sup_{f,g\in\mathcal{F}'}\left\{\frac{1}{n}\sum_{i=1}^{n}4\|\psi\|_{\mathrm{lip}}(f_i - g_i)\varepsilon_i' - \frac{\alpha|f_i - g_i|^p}{3^p}\right\}\right). \qquad (16)$$

*Proof.* We'll work forwards from (10), which says that

$$\mathbb{E}\Psi(\mathcal{E}(\tilde{f})) \leq \mathbb{E}\Psi\left(\sup_{f\in\mathcal{F}'}\left\{\frac{1}{n}\sum_{i=1}^{n}2\varepsilon_i'(\psi(f^*)_i - \psi(f)_i - \tfrac{1}{2}\mu(\tfrac{1}{3}d(f_i, f_i^*))) - \mu(\tfrac{1}{3}d(f_i, f_i^*))\right\}\right)$$

$$\leq \mathbb{E}\Psi\left(\sup_{f,g\in\mathcal{F}'}\left\{\frac{1}{n}\sum_{i=1}^{n}2\varepsilon_i'(\psi(g)_i - \psi(f)_i - \tfrac{1}{2}\mu(\tfrac{1}{3}d(f_i, g_i))) - \mu(\tfrac{1}{3}d(f_i, g_i))\right\}\right),$$

where the second step enlarges the domain in the supremum. For (15), we plug in the definition of the offset from Proposition 11

$$\mu(\tfrac{1}{3}d(x,y)) = \frac{|\psi(x) - \psi(y)|^2}{18m \vee 36/\eta}.$$

Under the condition that $\psi$ takes values in $[0, m]$, we have that

$$|\psi(x) - \psi(y)|^2 \leq 2m|\psi(x) - \psi(y)| \leq (2m \vee 4/\eta)|\psi(x) - \psi(y)|.$$

It follows that we can apply our "contraction lemma" for offset processes, Lemma G2, with the contractions

$$|\psi(x) - \psi(y) - \tfrac{1}{2}\mu(\tfrac{1}{3}d(x,y))|$$
$$\leq |\psi(x) - \psi(y)| + |\tfrac{1}{2}\mu(\tfrac{1}{3}d(x,y))|$$
$$\leq |\psi(x) - \psi(y)| + \frac{(m \vee 2/\eta)|\psi(x) - \psi(y)|}{18m \vee 36/\eta} \leq \frac{19}{18}|\psi(x) - \psi(y)|.$$

For (16), we first require the following lemma.

**Lemma C1.** *Let $\psi : \mathbb{R} \to \mathbb{R}$ be $(\mu, d)$-convex and $\|\psi\|_{\mathrm{lip}}$-Lipschitz with respect to $d(x, y) = |x - y|$. Let $r \geq 0$ be the largest constant such that $\mu(cx) \leq c^r\mu(x)$ for $c \leq 1$ which is non-negative by monotonicity of $\mu$. Then*

$$\mu(\tfrac{1}{3}|x - y|) \leq \left(\tfrac{2}{3}\right)^r \|\psi\|_{\mathrm{lip}}|x - y|.$$

Applying the lemma with $r = p$, we simply apply the contractions

$$|\psi(x) - \psi(y) - \tfrac{1}{2}\mu(\tfrac{1}{3}d(x,y))| \leq |\psi(x) - \psi(y)| + \tfrac{1}{2}\mu(\tfrac{1}{3}d(x,y)) \leq (1 + 2^{p-1}/3^p)\|\psi\|_{\mathrm{lip}}|x - y|$$

using Lemma G2, and then plug in

$$\mu(\tfrac{1}{3}d(x,y)) = \frac{\alpha|x - y|^p}{3^p},$$

from the definition of $p$-uniform convexity.

*Proof of Lemma C1.* Let $z$ be the minimizer of $\psi$ over $[x, y]$. By Lemma 3 we have
$$\psi(x) - \psi(z) \geq \mu(|x - z|), \qquad \psi(y) - \psi(z) \geq \mu(|y - z|). \tag{25}$$
Since $|x - y| = |x - z| + |y - z|$ because $z \in [x, y]$, we have
$$\mu(\tfrac{1}{3}|x - y|) \leq \mu(\tfrac{1}{3}(|x - z| + |y - z|)).$$
By monotonicity of $\mu$, this is
$$\leq \mu(\tfrac{2}{3}(|x - z| \vee |y - z|))$$
$$\leq \left(\tfrac{2}{3}\right)^r \{\mu(|x - z|) \vee \mu(|y - z|)\}.$$
Using (25), we have
$$\leq \left(\tfrac{2}{3}\right)^r |\psi(x) - \psi(z)| \vee \left(\tfrac{2}{3}\right)^r |\psi(y) - \psi(z)|$$
$$\leq \left(\tfrac{2}{3}\right)^r \|\psi\|_{\text{lip}} |x - z| \vee \left(\tfrac{2}{3}\right)^r \|\psi\|_{\text{lip}} |y - z|$$
$$\leq \left(\tfrac{2}{3}\right)^r \|\psi\|_{\text{lip}} |x - y|,$$
where in the last step we again used that
$$|x - y| = |x - z| + |y - z| \geq |x - z| \vee |y - z|$$
by our choice of $z$. This completes the proof. $\qquad\square$

$\square$

# D   Proof of Proposition 15

We recall Proposition 15.

**Proposition 15.** *If $\psi$ is a self-concordant loss and $\hat{f}$ is the empirical risk minimizer in a convex class $\mathcal{F}$, then*
$$\mathbb{E}\Psi(\mathcal{E}(\tilde{f})) \leq \mathbb{E}\Psi\left(\sup_{f \in \mathcal{F}'}\left\{\frac{1}{n}\sum_{i=1}^n 4(\psi_i(f) - \psi(f_i^*))\varepsilon_i' - \omega\left(\|f_i - f_i^*\|_{\psi, f_i^*}\right)\right\}\right), \tag{18}$$
*for $\omega(z) = z - \log(1 + z)$, $\|z\|_{\psi, w} \doteq \sqrt{z^2 \psi''(w)}$, and $(\varepsilon_i')_{i=1}^n$ are independent, symmetric Rademacher random variables and $\Psi$ is any increasing, convex function.*

*Proof.* Combining the self-concordance inequality Lemma 14 with Lemmas 3 and 5 immediately gives us
$$\mathcal{E}(\hat{f}) = \mathbb{E}\psi(\hat{f}) - \mathbb{E}\psi(f^*) \geq \mathbb{E}\omega(\|\hat{f} - f^*\|_{\psi, f^*}) \tag{26}$$
for the empirical risk minimizer $\hat{f}$ in a convex class. Since $\hat{f}$ is the risk minimizer, we also have $\mathbb{E}_n\psi(f^*) - \mathbb{E}_n\psi(\hat{f}) \geq 0$. Adding these two and rearranging, we have
$$\mathcal{E}(\hat{f}) \leq 2\mathcal{E}(\hat{f}) - \mathbb{E}\omega(\|\hat{f} - f^*\|_{\psi, f^*})$$
$$\leq 2(\mathbb{E} - \mathbb{E}_n)\psi(\hat{f}) - 2(\mathbb{E} - \mathbb{E}_n)\psi(f^*) - \mathbb{E}\omega(\|\hat{f} - f^*\|_{\psi, f^*})$$
$$\leq 2\sup_{f \in \mathcal{F}}\left\{(\mathbb{E} - \mathbb{E}_n)\psi(\hat{f}) - (\mathbb{E} - \mathbb{E}_n)\psi(f^*) - \tfrac{1}{2}\mathbb{E}\omega(\|f - f^*\|_{\psi, f^*})\right\}$$
Applying $\mathbb{E}\Psi$ on both sides gives
$$\mathbb{E}\Psi(\mathcal{E}(\hat{f})) \leq \mathbb{E}\Psi\left(2\sup_{f \in \mathcal{F}}\left\{(\mathbb{E} - \mathbb{E}_n)\psi(\hat{f}) - (\mathbb{E} - \mathbb{E}_n)\psi(f^*) - \tfrac{1}{2}\mathbb{E}\omega(\|f - f^*\|_{\psi, f^*})\right\}\right)$$
$$= \mathbb{E}\Psi\left(2\sup_{f \in \mathcal{F}}\left\{(\mathbb{E} - \mathbb{E}_n)(\psi(\hat{f}) - \psi(f^*)) - \tfrac{1}{4}(\mathbb{E} + \mathbb{E})\omega(\|f - f^*\|_{\psi, f^*})\right\}\right)$$
By Jensen's inequality, this is
$$\leq \mathbb{E}\Psi\left(2\sup_{f \in \mathcal{F}}\left\{(\mathbb{E} - \mathbb{E}_n)(\psi(\hat{f}) - \psi(f^*)) - \tfrac{1}{4}(1 + \mathbb{E})\omega(\|f - f^*\|_{\psi, f^*})\right\}\right).$$
The proof is then complete after applying Lemma G1 with $A(f) = 2(\psi(f) - \psi(f^*))$, $T = \mathcal{F}$, and $B(f) = \tfrac{1}{2}\omega(\|f - f^*\|_{\psi, f^*})$. $\qquad\square$

# E   Proof of Theorem 19

We recall Theorem 19.

**Theorem 19.** *Let $\psi$ be an $\eta$-exp-concave loss taking values in $[0, m]$. Then, with probability at least $1 - 9e^{-z}$, the star estimator $\tilde{f}$ applied to $(\psi, \mathcal{F})$ satisfies*

$$\mathcal{E}(\tilde{f}) \leq \inf_{0 \leq \alpha \leq \gamma} \left\{ 4\alpha + \frac{10}{\sqrt{n}} \int_\alpha^\gamma \sqrt{H_2(s)}\, ds + \frac{2H_2(\gamma)}{cn} + \frac{\gamma\sqrt{8\pi}}{\sqrt{n}} + \left( \frac{2}{cn} + \frac{\gamma\sqrt{8}}{\sqrt{n}} \right) z \right\}, \quad (23)$$

*where $H_2(s) \doteq H_2(s, \psi \circ \mathcal{F}')$ and $c = 36^{-1}(1/m \wedge \eta/2)$.*

*Proof.* We work forwards from (15), which tells us

$$\mathbb{E}\Psi(\mathcal{E}(\tilde{f})) \leq \mathbb{E}\Psi\left( \sup_{t \in T} \left\{ Z_t - cZ_t^2 \right\} \right),$$

where we define $Z$, $t$, $T$, and $c$ according to

$$Z_t = \frac{1}{n} \sum_{i=1}^n 4\varepsilon_i' t(X_i, Y_i) = \frac{1}{n} \sum_{i=1}^n 4\varepsilon_i'(\psi(f(X_i), Y_i) - \psi(g(X_i), Y_i)),$$

$$t = \psi(f(X_i), Y_i) - \psi(g(X_i), Y_i),$$

$$T = \psi \circ \mathcal{F}' - \psi \circ \mathcal{F}',$$

$$c = \frac{1}{36} \left( \frac{1}{m} \vee \frac{\eta}{2} \right).$$

Let $V$ be a covering of $T$ at resolution $\gamma$ in $L^2(\mathbb{P}_n)$ that is chosen to include 0, so that $\#V \leq \exp(2H_2(\gamma))$ almost surely by construction of $T$ and definition of $H_2(-)$. Then we can choose $\pi : T \to V$ with the properties that (1) $\|t - \pi(t)\|_{2,\mathbb{P}} \leq \gamma$ uniformly over $t \in T$, and (2) $\pi(t) = 0$ if $\|t\|_{2,\mathbb{P}} < \gamma$.

The proof will proceed in three lemmas which will be stated below and proved subsequently. The first lemma shows that $\sup_{t \in T} \left\{ Z_t - cZ_t^2 \right\}$ can be controlled in terms of (i) the local complexity of $(Z_t)_{t \in T}$ at scale $\gamma$ and (ii) the offset complexity of a finite approximation to $(Z_t)_{t \in T}$ at resolution $\gamma$. The second and third lemmas develop high-probability bounds for these two terms.

**Lemma E1** (from Liang et al. (2015, Lemma 6)). *It holds almost surely that*

$$\sup_{t \in T} \left\{ Z_t - cZ_t^2 \right\} \leq \sup_{t \in T} \left\{ Z_t - Z_{\pi(t)} \right\} + \sup_{v \in V} \left\{ Z_v - (c/4)Z_v^2 \right\}. \quad (27)$$

**Lemma E2.**

$$\mathbb{P}\left( \sup_{t \in T} \left\{ Z_t - Z_{\pi(t)} \right\} \geq 4\alpha + \frac{10}{\sqrt{n}} \int_\alpha^\gamma \sqrt{2H_2(s)}\, ds + \gamma\sqrt{\frac{8\pi}{n}} + x \right) \leq 2e^{-nx^2/(8\gamma^2)} \quad (28)$$

**Lemma E3.**

$$\mathbb{P}\left( \sup_{v \in V} \left\{ Z_v - (c/4)Z_v^2 \right\} > \frac{4H_2(\gamma) + 2x}{cn} \right) \leq e^{-x}. \quad (29)$$

Applying a union bound to the event in (28) with $x = z\gamma\sqrt{8}$, the event in (29) with $z = x$, and the complement of the event (27), we obtain that with probability at least $1 - 3e^{-z}$

$$\sup_{t \in T} \left\{ Z_t - Z_{\pi(t)} \right\} \leq 4\alpha + \frac{10}{\sqrt{n}} \int_\alpha^\gamma \sqrt{H_2(s)}\, ds + \frac{2H_2(\gamma)}{cn} + \frac{\gamma\sqrt{8\pi}}{\sqrt{n}} + \left( \frac{2}{cn} + \frac{\gamma\sqrt{8}}{\sqrt{n}} \right) z$$

Finally, since $\mathbb{E}\Psi(\mathcal{E}(\tilde{f})) \leq \mathbb{E}\Psi\left( \sup_{t \in T} \left\{ Z_t - cZ_t^2 \right\} \right)$ for all convex and increasing $\Psi$, we can apply the following.

**Lemma E4** ( Panchenko (2003, Lemma 1)). *If $\mathbb{E}\Psi(X) \leq \mathbb{E}\Psi(Y)$ for all convex and increasing functions $\Psi$, then*

$$\mathbb{P}(Y \geq t) \leq Ae^{-at} \implies \mathbb{P}(X \geq t) \leq Ae^{1-at}.$$

Thus, we have

$$\mathcal{E}(\tilde{f}) \le 4\alpha + \frac{10}{\sqrt{n}} \int_\alpha^\gamma \sqrt{H_2(s)} \, ds + \frac{2H_2(\gamma)}{cn} + \frac{\gamma\sqrt{8\pi}}{\sqrt{n}} + \left( \frac{2}{cn} + \frac{\gamma\sqrt{8}}{\sqrt{n}} \right) z$$

with probability at least $1 - (3e)e^{-z}$. After noting that $0 \le \alpha \le \gamma$ is arbitrary and $3e \le 9$, the proof is complete. $\qquad\square$

*Proof of Lemma E1.* We have

$$\sup_{t \in T} \left\{ Z_t - cZ_t^2 \right\} = \sup_{t \in T} \left\{ (Z_t - Z_{\pi(t)}) + ((c/4)Z_{\pi(t)}^2 - cZ_t^2) - \left( cZ_t^2 + Z_{\pi(t)} - (c/4)Z_{\pi(t)}^2 \right) \right\}$$

$$\le \sup_{t \in T} \left\{ Z_t - Z_{\pi(t)} \right\} + \sup_{v \in V} \left\{ Z_v - (c/4)Z_v^2 \right\},$$

provided we can show that the middle term $(c/4)Z_{\pi(t)}^2 - cZ_t^2$ is a.s. non-positive. To see this, note that either $\|t\|_{2,\mathbb{P}_n} < \gamma$, in which case by construction $\pi(t) = 0$ and $Z_{\pi(t)}^2 = 0$, so we are done, or else we have

$$\|\pi(t)\|_{2,\mathbb{P}_n} \le \|\pi(t) - t\|_{2,\mathbb{P}_n} + \|t\|_{2,\mathbb{P}_n} \le \|t\|_{2,\mathbb{P}_n} + \gamma \le 2\|t\|_{2,\mathbb{P}_n},$$

so that $\|\pi(t)\|_{2,\mathbb{P}_n}^2 \le 4\|t\|_{2,\mathbb{P}_n}^2$. But, after plugging in the definition of $Z_t$, the middle term is precisely

$$\frac{16c}{n} \left( \frac{\|\pi(t)\|_{2,\mathbb{P}_n}^2}{4} - \|t\|_{2,\mathbb{P}_n}^2 \right),$$

so we are done. $\qquad\square$

*Proof of Lemma E2.* Keeping in mind that $\|t - \pi(t)\|_{2,\mathbb{P}_n} \le \gamma$ and applying the chaining result in Srebro et al. (2010, Lemma A.3) gives us

$$\mathbb{E}_\varepsilon \sup_{t \in T} \left\{ Z_t - Z_{\pi(t)} \right\} \le 4\alpha + \frac{10}{\sqrt{n}} \int_\alpha^\gamma \sqrt{2H_2(s)} \, ds \tag{30}$$

almost surely with respect to the data, where we used that

$$\ln N(s, T, L^2(\mathbb{P}_n)) \le 2 \ln N(s, \psi \circ \mathcal{F}', L^2(\mathbb{P}_n)) \le 2H_2(s)$$

by definition of $T$ and the fact that $H_2(-)$ is an almost-sure bound on the logarithm of the $L^2(\mathbb{P}_n)$ covering numbers. It follows by applying Ledoux and Talagrand (1991, Theorem 4.7) with $\sigma^2(X) = \gamma^2/n$ that

$$\mathbb{P}_\epsilon \left( \sup_{t \in T} \left\{ Z_t - Z_{\pi(t)} \right\} \ge M_\epsilon + x \right) \le 2e^{-nx^2/(8\gamma^2)}, \tag{31}$$

where $\mathbb{P}_\varepsilon$ denotes the probability with respect to the multipliers $\varepsilon$ conditional upon the data and $M_\epsilon$ is a conditional median of $\sup_{t \in T} \left\{ Z_t - Z_{\pi(t)} \right\}$. Finally, we can deduce the upper bound

$$\mathbb{E}_\varepsilon \sup_{t \in T} \left\{ Z_t - Z_{\pi(t)} \right\} - M_\epsilon$$

$$\le \mathbb{E}_\epsilon \left[ \left( \sup_{t \in T} \left\{ Z_t - Z_{\pi(t)} \right\} - M_\epsilon \right) \mathbb{1} \left\{ \sup_{t \in T} \left\{ Z_t - Z_{\pi(t)} \right\} > M_\epsilon \right\} \right]$$

$$= \int_0^\infty \mathbb{P}_\epsilon \left( \sup_{t \in T} \left\{ Z_t - Z_{\pi(t)} \right\} - M_\epsilon > t \right) dt$$

$$\le \int_0^\infty 2e^{-nt^2/(8\gamma^2)} dt = \gamma \sqrt{\frac{8\pi}{n}} \tag{32}$$

Finally, putting together (30), (31) and (32) gives us

$$\mathbb{P}_\epsilon \left( \sup_{t \in T} \left\{ Z_t - Z_{\pi(t)} \right\} \ge 4\alpha + \frac{10}{\sqrt{n}} \int_\alpha^\gamma \sqrt{2H_2(s)} \, ds + \gamma \sqrt{\frac{8\pi}{n}} + x \right) \le 2e^{-nx^2/(8\gamma^2)}.$$

Since this conditional bound holds almost surely with respect to the data, we immediately deduce (28). $\qquad\square$

*Proof of Lemma E3.* Working conditionally upon the data, we can compute by applying Markov's inequality that

$$\mathbb{P}_\varepsilon \left( \sup_{v \in V} \left\{ Z_v - (c/4)Z_v^2 \right\} > t \right) = \mathbb{P}_\varepsilon \left( \exp\left( rn \sup_{v \in V} \left\{ Z_v - (c/4)Z_v^2 \right\} \right) > e^{rnt} \right)$$

$$\leq \mathbb{E}_\varepsilon \exp\left( rn \sup_{v \in V} \left\{ Z_v - (c/4)Z_v^2 \right\} \right) e^{-rnt}.$$

We can further compute that

$$\mathbb{E}_\varepsilon \exp\left( rn \sup_{v \in V} \left\{ Z_v - (c/4)Z_v^2 \right\} \right) = \mathbb{E}_\varepsilon \sup_{v \in V} \exp\left( \sum_{i=1}^n r\varepsilon_i' v_i - (c/4)rv_i^2 \right)$$

$$\leq \sum_{v \in V} \mathbb{E}_\varepsilon \exp\left( \sum_{i=1}^n \frac{r^2 v_i^2}{2} - \frac{crv_i^2}{4} \right),$$

by applying Hoeffding's lemma to each expectation with respect to the variables $\varepsilon_i$. Taking $r = c/2$, this is precisely $\#V$. Thus, we have that

$$\mathbb{P}_\varepsilon \left( \sup_{v \in V} \left\{ Z_v - (c/4)Z_v^2 \right\} > t \right) \leq \exp\left( \ln(\#V) - \frac{cnt}{2} \right).$$

Since $\ln(\#V) \leq 2H_2(\gamma)$ almost surely, we can deduce the unconditional bound

$$\mathbb{P} \left( \sup_{v \in V} \left\{ Z_v - (c/4)Z_v^2 \right\} > t \right) \leq \exp\left( 2H_2(\gamma) - \frac{cnt}{2} \right).$$

Taking $t = (4H_2(\gamma) + 2x)/cn$ gives (29). $\qquad\square$

# F  Proofs of Section 5 Results

## F.1  Proof of Corollary 16

We state Corollary 16.

**Corollary 16** (cf. Mendelson (2002, Theorem 5.1)). *Let $\psi(f, y) = |f - y|^p$ for $p > 1$ and let the class $\mathcal{F}$ and response $Y$ take values in $[-B, B]$. Then there exists a universal $C(p, B) = O(p2^p B^p)$ such that the star estimator $\tilde{f}$ has excess $\psi$-risk bounded as*

$$\mathbb{P} \left( \mathcal{E}(\tilde{f}, \mathcal{F}) \geq \epsilon + \frac{C(p, B)(H_2(\epsilon/C_{p,B}, \mathcal{F}) + \ln(1/\epsilon) + \ln(1/\rho))}{n} \right) \leq \rho, \tag{20}$$

*Proof.* This follows as a result of the more general bound (19), which says that

$$\mathbb{P} \left( \mathcal{E}(\tilde{f}) \geq \epsilon + \left( 36m \vee \frac{72}{\eta} \right) \left( \frac{H_2(\epsilon, \psi \circ \mathcal{F}') + \ln(1/\rho)}{n} \right) \right) \leq \rho.$$

In order to deduce (20), we need to bound the quantities $m$, $1/\eta$, and $H_2(\psi \circ \mathcal{F}')$. For $m$, since $|f|, |y| \leq B$, it must hold that $|f - y|^p \leq 2^p B^p$. For $\eta$, we can compute that

$$(\psi')^2/\psi'' = \frac{p^2 z^{2p-2}}{p(p-1)z^{p-2}} \leq \frac{pz^p}{p-1} \leq \frac{p2^p B^p}{p-1}$$

for $z = |f - y| \leq 2B$. Finally, we have $\|\psi\|_{\mathrm{lip}} \leq p2^p B^{p-1}$ by bounding the first derivative, so that we have the entropy estimates

$$H_2(\epsilon, \psi \circ \mathcal{F}') \leq H_2\left( \frac{\epsilon}{p2^p B^{p-1}}, \mathcal{F}' \right) \leq 2H_2\left( \frac{\epsilon}{p2^{p+1} B^{p-1}}, \mathcal{F} \right) + \ln\left( \frac{4B}{\epsilon} \right),$$

where the last step follows by applying Lemma G4 with $R \leq \sup_{f,y} |f - y| \leq 2B$. $\qquad\square$

### F.2 Proof of Lemma 17

We recall Lemma 17.

**Lemma 17** (Foster et al. (2018)). *For all $f$ and $\delta \in (0, 1/2]$, the excess risk relative to $\mathcal{L}_\delta$ satisfies*

$$\mathcal{E}(f; \mathcal{L}_\delta) \le \mathcal{E}(f; \mathcal{L}) + 2\delta$$

*Proof.* We can compute that

$$\ln(f) - \ln((1-\delta)f + \delta) = \ln\left(\frac{f}{(1-\delta)f + \delta}\right) \le \ln\left(\frac{1}{1-\delta}\right) \le 2\delta, \tag{33}$$

since $0 \le -\ln(1-\delta) \le 2\delta$ for $0 \le \delta \le 1/2$. Consequently, for any $g$,

$$\mathcal{E}(g; \mathcal{L}_\delta) = \mathbb{E}[-\ln g] - \inf_{f \in \mathcal{L}_\delta} \mathbb{E}[-\ln f]$$

$$= \mathbb{E}[-\ln g] - \inf_{f \in \mathcal{L}} \mathbb{E}[-\ln f] + \inf_{f \in \mathcal{L}} \mathbb{E}[-\ln f] - \inf_{f \in \mathcal{L}_\delta} \mathbb{E}[-\ln f]$$

$$= \mathcal{E}(g; \mathcal{L}) + \left(\inf_{f \in \mathcal{L}} \mathbb{E}[-\ln f] - \inf_{f \in \mathcal{L}_\delta} \mathbb{E}[-\ln f]\right)$$

By separability of the two infima, this is the same as

$$= \mathcal{E}(g; \mathcal{L}) + \sup_{h \in \mathcal{L}_\delta} \inf_{f \in \mathcal{L}} \mathbb{E}[\ln h - \ln f].$$

By choosing $h = (1-\delta)f + \delta$, the outer supremum may be bounded as

$$\ge \mathcal{E}(g; \mathcal{L}) + \inf_{f \in \mathcal{L}} \mathbb{E}[\ln((1-\delta)f + \delta) - \ln f]$$

$$\ge \mathcal{E}(g; \mathcal{L}) - 2\delta,$$

where the final step follows from negating (33). $\qquad\square$

### F.3 Proof of Corollary 18

We recall Corollary 18.

**Corollary 18.** *With probability at least $1 - \rho$, the star estimator $\tilde{f}_\delta$ in $\mathcal{L}_\delta$ satisfies*

$$\mathcal{E}(\tilde{f}_\delta; \mathcal{L}) \le \epsilon + 2\delta + C \ln(1/\delta)\left(\frac{H_2(\delta\epsilon, \mathcal{L}) + \ln(1/\epsilon\delta) + \ln(1/\rho)}{n}\right) \tag{21}$$

*Let $\mathcal{L}$ be the generalized linear model corresponding to*

$$\mathcal{F}_B = \left\{x \mapsto Wx \,\middle|\, W \in \mathbb{R}^{k \times q}, \|W\|_{2 \to \infty} \le B\right\}$$

*with A-Lipschitz, surjective link $\varphi$ and features $X \in \mathbb{R}^q$ that satisfy $\|X\|_2 \le R\sqrt{q}$. Then with probability at least $1 - \rho$*

$$\mathcal{E}(\varphi^\dagger \circ \tilde{f}_\delta; \mathcal{F}_B) \le \frac{\ln(n)}{n}\left\{Ckq \ln(ABRn\sqrt{k}) + \ln(1/\rho)\right\}. \tag{22}$$

*Proof.* By Lemma 17, it suffices for (21) to show instead that

$$\mathbb{P}\left(\mathcal{E}(\tilde{f}_\delta; \mathcal{L}_\delta) > \epsilon + C \ln(1/\delta)\left(\frac{H_2(\delta\epsilon, \mathcal{L}) + \ln(1/\epsilon\delta) + \ln(1/\rho)}{n}\right)\right) \le \rho.$$

This in turn follows from the general inequality (19), which says in this context that

$$\mathbb{P}\left(\mathcal{E}(\tilde{f}_\delta, \mathcal{L}_\delta) \ge \epsilon + \left(36m \vee \frac{72}{\eta}\right)\left(\frac{H_2(\epsilon, -\ln \circ \mathcal{L}'_\delta) + \ln(1/\rho)}{n}\right)\right) \le \rho.$$

Since $\mathcal{L}'_\delta$ takes values in $[\delta, 1]$, the log loss takes values in $[0, \ln(1/\delta)]$, so we choose $m = \ln(1/\delta)$. Since the log loss is 1-exp-concave, we choose $\eta = 1$. Finally, the log loss in this domain is $(1/\delta)$-Lipschitz, so we have the estimates

$$H_2(\epsilon, -\ln \circ \mathcal{L}'_\delta) \le H_2(\delta\epsilon, \mathcal{L}'_\delta) \le 2H_2(\delta\epsilon/2, \mathcal{L}_\delta) + \ln(2\ln(1/\delta)/\delta\epsilon),$$

where the last step follows from Lemma G4 with $R = \ln(1/\delta)$. Finally, use that $H_2(-, \mathcal{L}_\delta) \leq H_2(-, \mathcal{L})$ since $\mathcal{L}_\delta$ is the image of $\mathcal{L}$ under a pointwise contraction, and simplify (for example absorbing $\ln \ln(1/\delta) \leq \ln(1/\delta)$) into the constant $C$). For (18), we plug in the covering estimates

$$H_2(\delta\epsilon, \mathcal{L}) \leq H_2\left(\frac{\delta\epsilon}{A}, \mathcal{F}_B\right) \leq \ln\left(\frac{ABR\sqrt{k}}{\epsilon\delta}\right)^{kd} = kd \ln\left(\frac{ABR\sqrt{k}}{\delta\epsilon}\right)$$

for $\mathcal{F}_B$, which are standard. Finally, we take $\epsilon = \delta = 1/n$ and simplify. $\qquad\square$

### F.4 Proof of Corollary 20

We recall Corollary 20.

**Corollary 20.** *Consider a generalized linear model with $A$-Lipschitz loss $f \mapsto -\ln\langle\varphi(f), y\rangle$. Suppose the entropy numbers $H_2(\epsilon; \mathcal{F})$ are of order $\epsilon^{-q}$. Then, the regularized star estimator $\varphi^\dagger \circ \tilde{f}_\delta$ with $\delta = 1/n$ satisfies the rates appearing on the left.*

*On the other hand, for an arbitrary class $\mathcal{L}$ taking values in $[0,1]$ subject to the log loss, the regularized star estimator $\tilde{f}'_{\delta'}$—for appropriately chosen $\delta'$—attains the rates appearing on the right.*

*Here, the symbol $\lesssim_\rho$ denotes an upper bound that holds with probability $1 - \rho$, hiding universal constants and a multiplicative factor $\ln(1/\rho)$.*

$$\mathcal{E}(\varphi^\dagger \circ \tilde{f}_\delta) \lesssim_\rho \begin{cases} A^q n^{-2/(2+q)}\ln(n) & q < 2 \\ A^q n^{-1/2}\ln(n) & q = 2 \\ A^q n^{-1/q} & q > 2 \end{cases} \qquad \mathcal{E}(\tilde{f}'_{\delta'}) \lesssim_\rho \begin{cases} n^{-1/(1+3q/2)} & q < 2 \\ n^{-1/4}\ln(n) & q = 2 \\ n^{-1/(2q)} & q > 2 \end{cases} \qquad (24)$$

*Proof.* These bounds are all derived by applying Theorem 19 under different assumptions on the entropy function. In particular combining (23) with Lemma 17—using the fact that the log loss over $\mathcal{L}_\delta$ takes values in $[0, \ln(1/\delta)]$ and is 1-exp-concave—gives us that with probability $1 - \rho$,

$$\mathcal{E}(\tilde{f}) \lesssim_\rho 2\delta + \inf_{0 \leq \alpha \leq \gamma} \left\{ 4\alpha + \frac{10}{\sqrt{n}} \int_\alpha^\gamma \sqrt{H_2(s)}\, ds + \frac{H_2(\gamma)\ln(1/\delta)}{n} + \frac{\gamma}{\sqrt{n}} \right\},$$

where the symbol $\lesssim_\rho$ hides universal constants and a multiplicative factor $\ln(1/\rho)$.

For the left-hand side results, the entropy numbers scale as $(A/\epsilon)^q$; choosing $\delta = \frac{1}{n}$, we get

$$\frac{2}{n} + 4\alpha + \frac{10A^{q/2}}{\sqrt{n}} \int_\alpha^\gamma s^{-q/2}\, ds + \frac{\gamma^{-q}A^q \ln n}{n} + \frac{\gamma}{\sqrt{n}}.$$

For the $q < 2$ case we take $\alpha = 0$ and $\gamma = n^{-1/(2+q)}$. For the case $q = 2$ we take $\alpha = 1/n$ and $\gamma = 1$. For the case $q > 2$ we take $\alpha = n^{-1/q}$ and $\gamma = 1$. For the right-hand side results, the entropy numbers scale as $(1/\delta\epsilon)^q$, giving us the bound

$$2\delta + 4\alpha + \frac{12\delta^{-q/2}}{\sqrt{n}} \int_\alpha^\gamma s^{-q/2}\, ds + \frac{\gamma^{-q}\delta^{-q}}{n} + \frac{\gamma}{\sqrt{n}}.$$

For $q < 2$, we take $\alpha = 0$, $\delta = n^{-1/(1+3q/2)}$, and $\gamma = n^{-1/(2+3q)}$. For $q = 2$ we take $\delta = n^{-1/4}$, $\alpha = 1/n$, and $\gamma = 1$. For $q > 2$ we take $\delta = \alpha = n^{-1/2p}$ and $\gamma = 1$. $\qquad\square$

## G Technical Lemmas

**Lemma G1** (Offset symmetrization). *For every increasing and convex function $\Psi$,*

$$\mathbb{E}\Psi\left(\sup_{t \in T}\left\{\frac{1}{n}\sum_{i=1}^n (1 - \mathbb{E})A_i(t) - (1 + \mathbb{E})B_i(t)\right\}\right) \leq \mathbb{E}\Psi\left(2\sup_{t \in T}\left\{\frac{1}{n}\sum_{i=1}^n \varepsilon'_i A_i(t) - B_i(t)\right\}\right).$$

*Proof.* Noting that $\mathbb{E}A_i(t) = \mathbb{E}A_i'(t)$ and $\mathbb{E}B_i(t) = \mathbb{E}B_i'(t)$, where $A_i'$ (respectively, $B_i$) is an independent copy of $A_i$ (resp. $B_i'$), and finally moving the expectations outside by applying Jensen's inequality to the convex function $\Psi(\sup_{t \in T}(-))$, we have

$$\mathbb{E}\Psi\left(\sup_{t \in T}\left\{\frac{1}{n}\sum_{i=1}^{n}(1-\mathbb{E})A_i(t) - (1+\mathbb{E})B_i(t)\right\}\right)$$
$$\leq \mathbb{E}\Psi\left(\sup_{t \in T}\left\{\frac{1}{n}\sum_{i=1}^{n}A_i(t) - A_i'(t) - B_i(t) - B_i'(t)\right\}\right).$$

Since $A_i - A_i'$ is symmetric, it is equal in distribution to $\varepsilon_i'(A_i - A_i')$, where $\varepsilon_i'$ is a symmetric Rademacher r.v. independent of $(A, A', B, B')$, hence we can write

$$= \mathbb{E}\Psi\left(\sup_{t \in T}\left\{\frac{1}{n}\sum_{i=1}^{n}\varepsilon_i'(A_i(t) - A_i'(t)) - B_i(t) - B_i'(t)\right\}\right)$$
$$= \mathbb{E}\Psi\left(\sup_{t \in T}\left\{\frac{1}{n}\sum_{i=1}^{n}\varepsilon_i'A_i(t) - B_i(t) + \frac{1}{n}\sum_{i=1}^{n}(-\varepsilon_i')A_i'(t) - B_i'(t)\right\}\right).$$
$$= \mathbb{E}\Psi\left(\sup_{t \in T}\left\{2\mathbb{E}_\sigma\left[\frac{\sigma}{n}\sum_{i=1}^{n}\varepsilon_i'A_i(t) - B_i(t) + \frac{1-\sigma}{n}\sum_{i=1}^{n}(-\varepsilon_i')A_i'(t) - B_i'(t)\right]\right\}\right),$$

where $\sigma$ is an independent symmetric Bernoulli r.v. By a final application of Jensen's inequality and equality of the distributions of $(\sigma\varepsilon_i')_{i=1}^{n}$ and $((1-\sigma)(-\varepsilon_i'))_{i=1}^{n}$, this is

$$\leq \mathbb{E}\Psi\left(2\sup_{t \in T}\left\{\frac{\sigma}{n}\sum_{i=1}^{n}\varepsilon_i'A_i(t) - B_i(t) + \frac{1-\sigma}{n}\sum_{i=1}^{n}(-\varepsilon_i')A_i'(t) - B_i'(t)\right\}\right),$$
$$= \mathbb{E}\Psi\left(2\sup_{t \in T}\left\{\frac{1}{n}\sum_{i=1}^{n}\varepsilon_i'A_i(t) - B_i(t))\right\}\right), \tag{34}$$

which is what we aimed to show. $\qquad\square$

**Lemma G2** (Offset contraction). *Suppose that $|A_i(s) - A_i(t)| \leq |C_i(s) - C_i(t)|$ for all $s, t \in T$. Then, for all increasing and convex $\Psi$, we have*

$$\mathbb{E}\Psi\left(2\sup_{t \in T}\left\{\frac{1}{n}\sum_{i=1}^{n}\varepsilon_i'A_i(t) - B_i(t)\right\}\right) \leq \mathbb{E}\Psi\left(2\sup_{t \in T}\left\{\frac{1}{n}\sum_{i=1}^{n}\varepsilon_i'C_i(t) - B_i(t)\right\}\right), \tag{35}$$

*whenever the $\varepsilon_i'$ are symmetric Rademacher variables that are independent of A, B and C.*

*Proof.* To simplify notation, put

$$S_m(t) = \sum_{i=1}^{m}\varepsilon_i'A_i(t) - B_i(t).$$

Writing out the expectation with respect to $\varepsilon_n'$ gives

$$\mathbb{E}\Psi\left(2\sup_{t \in T}\left\{\frac{1}{n}\sum_{i=1}^{n}\varepsilon_i'A_i(t) - B_i(t)\right\}\right)$$
$$= \mathbb{E}\Psi\left(\frac{1}{n}\sum_{j=1}^{2}\sup_{t \in T}\left\{(-1)^j A_n(t) + S_{n-1}(t) - B_n(t)\right\}\right)$$
$$= \mathbb{E}\Psi\left(\frac{1}{n}\sup_{s,t \in T}\left\{A_n(s) - A_n(t) + (S_{n-1}(s) - B_n(s)) - (S_{n-1}(t) - B_n(t))\right\}\right).$$

Applying our assumption that $|A_i(s) - A_i(t)| \leq |C_i(s) - C_i(t)|$, this is

$$\leq \mathbb{E}\Psi\left(\frac{1}{n}\sup_{s,t \in T}\left\{|C_n(s) - C_n(t)| + (S_{n-1}(s) - B_n(s)) - (S_{n-1}(t) - B_n(t))\right\}\right).$$

Since the argument of the supremum is symmetric in $(s, t)$, we can remove the absolute value, yielding

$$\leq \mathbb{E}\Psi\left(\frac{1}{n}\sup_{s,t\in T}\left\{C_n(s) - C_n(t) + (S_{n-1}(s) - B_n(s)) + (S_{n-1}(t) - B_n(t))\right\}\right).$$

Since the supremum is now separable in $(s, t)$, we further have

$$= \mathbb{E}\Psi\left(\frac{1}{n}\sum_{j=1}^{2}\sup_{t\in T}\left\{(-1)^j C_n(t) - B_n(t) + S_{n-1}(t)\right\}\right)$$

$$= \mathbb{E}\Psi\left(\frac{2}{n}\sup_{t\in T}\left\{\varepsilon_n C_n(t) - B_n(t) + S_{n-1}(t)\right\}\right).$$

Applying these manipulations to each summand $r$ from $n-1$ down to $1$ gives us

$$\leq \mathbb{E}\Psi\left(\sup_{t\in T}\left\{\frac{2}{n}\sum_{i=r}^{n}\varepsilon_i C_i(t) - B_i(t) + S_{r-1}(t)\right\}\right)$$

$$\leq \mathbb{E}\Psi\left(2\sup_{t\in T}\left\{\frac{1}{n}\sum_{i=1}^{n}\varepsilon_i C_i(t) - B_i(t)\right\}\right),$$

which is what we aimed to show. $\qquad\square$

**Lemma G3** (Log margin computation). *For $|z| \leq c$,*

$$e^{-z} + z - 1 \geq \frac{z^2}{2c \vee 4}.$$

*Proof.* Note that $z - 1 \geq z/2 \geq z^2/(2c)$ for $z \geq 2$. So it suffices to check the inequality for $z < 2$. On the other hand, one can check by minimizing the left-hand side that

$$\frac{e^{-z} + z - 1}{z^2} \geq \frac{1}{4}$$

for $0 < z < 2$ (the derivative of the left-hand side is negative, and the inequality holds at $z = 2$). Finally, the inequality for $z \leq 0$ follows by noting that

$$e^{-z} - 1 + z = \frac{z^2}{2} + \sum_{k=3}^{\infty}\frac{(-z)^k}{k!},$$

by the series expansion for $e^{-z}$ and the remainder term must be non-negative for $z \leq 0$. $\qquad\square$

**Lemma G4** (cf. Mendelson (2002, Lemma 4.5)). *Put $\mathcal{F}' = \cup_{\lambda\in[0,1]}\lambda\mathcal{F} + (1-\lambda)\mathcal{F}$ and $R_\mu = \sup_{f\in\mathcal{F}}\|f\|_{L^2(\mu)}$. Let $N_2(\epsilon, S, \mu)$ denote the $\epsilon$-covering number of the set $S$ in $L^2(\mu)$. Then*

$$N_2(\epsilon, \mathcal{F}', \mu) \leq \left(\frac{2R_\mu}{\epsilon}\right)N_2(\epsilon/2, \mathcal{F}, \mu)^2. \tag{36}$$

*Consequently, if $R = \sup_\mu R_\mu$ where the supremum is over probability measures,*

$$H_2(\epsilon, \mathcal{F}') \leq 2H_2(\epsilon/2, \mathcal{F}) + \ln\left(\frac{2R}{\epsilon}\right). \tag{37}$$

*Proof.* Let $S$ denote a minimal covering of $\mathcal{F}$ in $L^2(\mu)$ at resolution $\epsilon/2$. Given some $(s, t) \in S^2$, let $T(s, t)$ denote an $\epsilon/2$ covering of the line segment interpolating $s$ and $t$. This line segment has length at most $2R_\mu$ in $L^2(\mu)$, hence $\#T(s, t) \leq \frac{2R_\mu}{\epsilon}$. We are therefore done if we can show that

$$\bigcup_{(s,t)\in S^2} T(s, t)$$

is an $\epsilon$ covering of $\mathcal{F}'$.

To this end, let $f \in \mathcal{F}'$ be given. By definition, we may write $f = \lambda f_1 + (1-\lambda)f_2$ for $f_1, f_2 \in \mathcal{F}$, and we can choose $s_1, s_2 \in S$ such that

$$\|s_1 - f_1\|_{L^2(\mu)}, \|s_2 - f_2\|_{L^2(\mu)} \leq \epsilon/2.$$

Due to convexity of the norm, we must have that

$$\|(\lambda s_1 + (1-\lambda)s_2) - f\|_{L^2(\mu)} \leq \epsilon/2.$$

By construction, there exists some $h \in T(s_1, s_2)$ such that

$$\|(\lambda s_1 + (1-\lambda)s_2) - h\|_{L^2(\mu)} \leq \epsilon/2.$$

Using the triangle inequality, we deduce $\|f - h\|_{L^2(\mu)} \leq \epsilon$ and the proof of (36) is complete; (37) then follows by first taking logarithms, then taking the supremum over probability measures $\mu$. $\qquad\square$