# OpenReview forum: "Localization, Convexity, and Star Aggregation"
_NeurIPS.cc/2021/Conference — NeurIPS 2021 Poster_

### Official Review · Reviewer_zjpm · 2021-07-16

**Rating:** 6
**Confidence:** 4

**Summary:**

This paper introduces a notion of $(\mu, d)$ convexity, with connections to strong convexity and self-concordancy, and shows that the excess risk is bounded by an offset Rademacher average, where the offset involves this convexity concept.
The authors show that their method is sufficient to handle a broad class of loss functions, including the $p$-loss and cross-entropy loss, generalizing previous work with square loss, and even operates on non-convex model spaces via the star algorithm. Finally, they show that for exponentially concave, uniformly convex, and self-concordant loss functions, their results have particularly convenient and elegant forms. In particular, they show relevant and interesting applications, with $p$-loss, and generalized linear models (improper maximum likelihood).


**Limitations And Societal Impact:**

Largely N/A

**Main Review:**

This paper was fascinating to read, and I am quite excited about the results!  Combining the star algorithm with offset Rademacher averages is clever, and the consequences are quite desirable, giving new insights into both. Some inconsistencies in the paper and confusing or underspecified notation mar readability and my understanding of the work slightly, but overall do not severely hinder the paper, though I would like to see them addressed. I have some technical questions regarding proposition 7, and concerns over its usability given that $f^*$ is not in general known, however these may be straightforward to address (upon which I would likely evaluate the paper more favorably).  I would also like to see more explicit comparisons to existing methods (in particular bound decay rates on common problems).




Item 1 of the contributions in the introduction states that you show that offset Rademacher averages work with $(\mu, d)$ convex functions, but proposition 7 is already using the star-algorithm approach. Presumably a similar result holds in the convex case, without the $\frac{1}{3}$ constant, but this would be nice to see in the paper, with a clear comparison to Liang 2015.



Most of my comments revolve around proposition 7.  It is a very nice result, however it does not match the statement in the appendix.  (9) in the appendix differs from (9) in the paper body, and the difference is somewhat non-trivial, as $\mu$ is multiplied by the Rademacher random variables in the appendix, but not in the body. It also looks like a 2 or a 4 factor might be missing from the $\mu$ term, if the offset term does not have a complementary constant factor.
In the appendix form, subtracting $\mu$ before multiplying by $\epsilon_i$ could severely decentralize the function family.  Could this be combined with (empirical) centralization arguments (see, e.g., [1]), to decrease the Rademacher average?



An obvious issue with applying this result is that it depends on the population-optimal $f^*$ which in machine learning applications, obviously we can't assume to know. It doesn't seem to me like replacing it with $\hat{f}$ could work without introducing severe bias that would be difficult to correct for (unlike with localized Rademacher averages, where empirical variances can play the role of true variances, with modest correction factors). It seems that in theorem 11, you fix this by worst-casing over $f^*$ (and calling it $g$), but I would prefer to see this described around proposition 7.  I’d also like to see if the supremum over $g$ could be moved outside the expectation, and if it could be over a subset of $\cal F$ that is near-optimal (i.e., that contains $f^{*}$ w.h.p.).

Another point I'd like to make on line 120 / proposition 5.  As far as I can tell, nothing is stopping you from applying the triangle inequality (in $d$) and Jensen’s inequality (in $\mu$) to get  $f(x) - f(\tilde{x}) \geq 3\mu(\frac{1}{3} d(x, \tilde{x}))$. This propagates throughout the paper to a significant improvement to your results, e.g., to proposition 7, and for $(p=2)$-uniform convexity (example 1), it improves your constant-factor gap from 9 to 3.  Faster-decaying offsets may also enable analysis on models for which bounds were previously vacuous.


Specifics:


39: “leading to fast upper bound” confusing, should be "a fast-rate" or "a fast-decaying"?

Proposition 1: It was not at all clear on an initial read what $k$ is.  Specifying $k$-class logistic regression would clarify significantly.  Furthermore, what good is a bound on the norm of the predictors, without also bounding the norm of the data? Is $B$ a bound on the dot product of data points and weights vectors?
Furthermore, in most of the paper, $d$ is a metric, and $B$ a vector space, this could get very confusing.

Definition 2: The subgradient is in general set-valued. I presume the definition needs to hold either for all or for some subgradient?  Furthermore, the condition is undefined for coincident $x, y$. $\mu$  needs to be extended, presumably such that $\mu(0) = 0$ and $\mu(x) > 0 \forall x > 0$, or you must restrict $x \neq y$?  Did you need to use, specifically, the sub-gradient anywhere?  Perhaps the paper would be better off stated for differentiable functions, with an explanation of how it may be extended to the subdifferentiable case?

From the proof of Lemma 4, I gather that it's for some subgradient rather than for all, as I believe that proof would fail, for instance, with the function $f(x) = x$ for some (but not all) subgradients.

Before lemma 6, some explanation of the metric $d_n$ would be greatly appreciated.


lemma 6: As far as I can tell, $x$ is a model output, not a model domain member, but this is confusing.  $\psi$ must be convex in what, the parameter space (function space), or $x$ itself?  It is also not clear what $f_{i}$ is (or $g_{i}$ later in the paper); presumably $f_i = \psi(f(X_i), Y_i)$?.  Either way, some type qualifiers on $x,y,\psi$, and $d$ would make such confusion impossible.

I'm a bit confused by the calculus notation on 153: is the limit an upper limit, or does the notation reflect some matter of converting the set-valued subgradient to a scalar?  And presumably the differentiation expresses a partial derivative, but $d$ (particularly when set in italics, rather than roman) is confusing, as it’s often used as a distance function?

156: Stating or bounding the constant would be appreciated.

Equation 7: It's not illegible, but some brackets on the expectations would not go amiss.

161: is $\equiv$ congruence / equivalence, or definition (sometimes written $\doteq$, $:=$, $=:$, or $=$ with a triangle)?


Prop 7: Eq. (9) of the appendix would be easier to read with variably-sized parentheses. Similar applies throughout the appendix.

Remark 8: Is this nomenclature standard?  I think of localization as inherently a scaling or second moment normalization process, whereas this is clearly a shifting argument.  This reviewer is admittedly more familiar with localized Rademacher averages than offset Rademacher averages; perhaps there is no issue.

208: Can $\sup_{g}$ go outside the expectation?

237: “While most general bound uses chaining and” -> “While the most general bound uses chaining, and”

239: Why do we need to worst-case over probability measures in a chaining bound?  Can we not use empirical covering numbers?  Also, calling the probability measure $\mu$ will lead to confusion with the function $\mu$.

317: “Jean-yves Audibert” -> “Jean-Yves Audibert”


References:

[1] "Sharp uniform convergence bounds through empirical centralization," C Cousins, M Riondato

**Time Spent Reviewing:**

9

---

> ### Author Response · Authors · 2021-08-07
> **Response to Reviewer jzpm**
>
> Thank you for your extensive feedback! We’re thrilled to hear you enjoyed the paper and think your suggestions will make it much stronger.
>
> ---
>
> ### ERM in convex classes
> Regarding our claim in item 1 of the contributions, it is remarked at lines 149-150 that in a convex class the star estimator coincides with regular ERM (this is because $\mathrm{star}(\hat f, \mathcal F) = \mathcal F$ in this case).
>
> Thus, our results automatically apply to convex ERM (although the constants can be improved by a factor of $9$, or $3^p$ under $p$-uniform convexity). We propose to emphasize this point as a self-standing “Remark.”
>
> ---
>
> ### Limitations of Prop. 7
>
> 1.  We first would like to clarify that **the result should appear as it does in the Appendix.** The Appendix version is the the result used downstream in the paper. We apologize for the typesetting mishap.
>
>     The presence of the $\varepsilon_i\mu(d(f_i,f^*_i))/2$ term does weaken the bound, although fortunately our proofs absorb this term during the contraction step at the expense of a small constant.
>
> 2. Although data-dependent bounds are not our focus, we believe using centralization is a very interesting possibility for tightening such bounds in practice. Data-dependent confidence bounds could be obtained by combining Eqn. (6) with a more complicated “high-probability” symmetrization argument, see Liang et. al. (2015), and we leave this to future work.
>
>     To elaborate on centralization in our context, the offset term ensures that w.h.p. the process in Eqn. (8) is negative outside of a small ball around the true optimum.
>
>     This ball's radius generally coincides with the usual “localization radius” or fixed point from the local Rademacher complexity argument, which is why it is common to refer to the phenomenon as “localization.”
>
>     Since we are interested in the fluctuations of the process in a small neighborhood around $f^*$ relative to its value at $f^*$, our object of interest is the *modulus of continuity* of the process, not the supremum.
>   It seems to us like the role centralization can play in bounding the modulus of continuity requires further investigation.
>
> 3. It is absolutely correct that we do not have access to $f^*$. Like you said, to use the result off-the-shelf one must take the supremum over possible $f^*$, as we do in Theorem 11. This can indeed be refined by taking the supremum outside the expectation.
>
>     As you suggest, one could also incorporate additional information about the location of $f^*$ to tighten the downstream bounds, and that is the primary reason why we stated a version of the result with $f^*$.
>
>     However, since our main motivation was to study the fast rate phenomenon and not to develop data-dependent confidence bounds, and since these improvements translate to at most a constant factor of 2 improvement in our final rates, we decided not to develop this further in the paper.
>  (The factor of 2 comes from the fact that the entropy satisfies $H_\epsilon(\mathcal F - \mathcal F) \lesssim 2 H_\epsilon(\mathcal F)$)
>
>     We agree that these limitations and possible extensions comprise an important discussion that should follow Prop. 7 and we will make room to include it in the paper.
>
> ---
> ### Empirical entropies
> Focusing on fast rates as opposed to data-dependent bounds is also why we pass immediately to the supremum over all probability measures in the chaining bound. All results should go through using the empirical entropy and working conditional upon the data, but with more complicated proofs.
>
> ---
>
> ### Tightening Prop. 5
> In regards to Prop. 5, we are having trouble seeing how such an improvement is possible. Our geometric argument shows that there exist points $s$ and $t$ such that $$f(x) - f(y) \ge \mu(d(x,s)) \vee \mu(d(s,t)) \vee  \mu(d(t,y)).$$ In the worst case, all three distances on the right are equal to $d(x,y)/3$ which recovers our inequality.
>
> Taking a weighted average of the three terms on the right, using Jensen’s inequality, and then optimizing over the weights gives the same result.
>
> The inequality you described would follow from $f(x) - f(y) \ge \mu(d(x,y))$ but unfortunately that stronger inequality only holds in a convex class.
>
> If we’re missing something, please let us know and we’ll happily propagate the improved constants through the paper.
>
> ---
> ### Sub-gradient
>
>  Thanks for calling our attention to this. We agree that it's better to state results only for differentiable losses and are happy to make this edit. Fortunately, we never require the generalization to arbitrary convex functions.
>
> For correctness and clarification, the condition in Def. 2 should be required to hold for all values in the sub-gradient. Then, it does not create problems that the inequality under line 105 in the proof of Lemma 4 holds only for a particular nonempty subset of the sub gradient, as you point out.
>
> ---
> Thank you also for the notational and writing edits, which we will absolutely incorporate.

---

> > ### Comment · Reviewer_zjpm · 2021-08-13
> > **Thank You**
> >
> > Thank you for your detailed response.
> >
> > That explanation of proposition 5 and the geometric argument as presented above is quite clear it might be nice to see some variant of it in the paper or appendix.
> > I had previously thought the maximum was actually a sum, in which case my argument would work, but if it is indeed the maximum over the three segments, then I don't see how the argument would hold.
> >
> > Speaking more coarsely, is this because we are traveling in different directions going from $x$ to $t_{1}$ and $s_1$ to $\tilde{x}$ than from $t_1$ to $s_1$ (“down”, “up”, “down”?)? If so, could it maybe improve to $\frac{2}{3} \mu(\frac{1}{3} d(x, \tilde{x}))$?

---

> > > ### Author Response · Authors · 2021-08-16
> > > **Improving Prop. 5**
> > >
> > > Yes, the problem is that the segments connecting $\partial f_{\tilde x}$ to $\partial f_x$ do not form a straight line.
> > >
> > > We see your point that if we do not yet apply the triangle inequality, we would get the modulus $$\mu((d(x,t_1)+d(t_1,s_1)+d(s_1,\tilde x))/3),$$ where the argument of $\mu$ is a tighter lower bound on the longest segment. So we can improve our bound when the three segments are very far from forming a direct path so that the triangle inequality is very loose.
> > >
> > > Meanwhile, my (non-rigorous!) intuition from looking at the figure is that when the segments *do* almost form a direct path it should be the case that a large fraction of the total length of path comes from the $d(x,t_1)$ term, which would also allow to improve the bound.
> > >
> > > Such an argument could in principle tighten the bound. However, if 1/9 is the optimal constant for the square loss as suggested by reviewer TjwQ (we were also unable to verify this claim) then there is no hope of improving our bound in this general setting, and the above intuition must fail.

---

### Official Review · Reviewer_TjwQ · 2021-07-16

**Rating:** 8
**Confidence:** 3

**Summary:**

This paper is concerned with the excess risk analysis of empirical risk minimization (ERM) over convex function classes, as well as Audibert's 'improper' Star-shaped aggregation procedure over nonconvex classes (which reduces to ERM in the convex setting), in statistical learning.
Sharp bounds have been obtained in the case of least-squares regression, and this paper considers additional settings such as exp-concave statistical learning, as well as losses satisfying different types of uniform convexity (such as p-convexity).
From a technical point of view, the analysis relies on an extension of the offset Rademacher complexities (introduced and leveraged for square loss by Liang, Rakhlin and Sridharan to analyze convex ERM and nonconvex Star) to other uniformly convex losses, in particular exp-concave ones; it is shown that an analogue of this offset process also controls the excess risk in these cases.
The main outcomes of this analysis are applications to regression with p-loss for p>2, and to improper learning of generalized linear models (such as logistic regression).


**Limitations And Societal Impact:**

Yes

**Main Review:**

This paper is clear and well-written, and contains a clean extension of results on Star aggregation for square loss to other standard settings such as exp-concave statistical learning. As far as I know, the extension of Star to the exp-concave setting (or for L^p regression), for instance, has not appeared before in the literature (different procedures were considered e.g. by Mehta).
I also appreciate the derivation of the negative quadratic term for exp-concave losses (reducing first to log-loss) in the right distance in Sec 4.
As a result, I definitely recommend acceptance of this paper at Neurips (I did not check the proofs enough to be able to confidently put a high.

Due to significant reviewing load and limited time, I did not really check the proofs in the supplementary material, so (as far as I am concerned) correctness is left to the authors' responsibility.
However, at a high level, I believe the authors' approach is sound. Indeed, Audibert's insight was that ERM can fail in the nonconvex setting if two functions are far from each other but have near-identical risk; however, in this case, the midpoint of the two will have better risk by the parallelogram identity. However, the argument is not specific to the negative quadratic term arising from the parallelogram identity, and it should also extend to other types of uniform convexity, as shown in the paper. Such moduli of convexity are in fact derived in detail both for the considered special cases in the main text, so the ingredients are in place.

Below are some additional comments:
- The abstract states: "Surprisingly, this condition is shown to capture exponential concavity and self-concordance, uniting several apparently disparate results."
As stated, this seems to be somewhat excessive, since unlike for exp-concave or p-loss, no explicit bound is stated for convex self-concordant ERM from the generic bound (2) as noted in lines 231-2, whether the bound can be leveraged in the improper setting is unclear due to the local nature of the quadratic approximation provided by self-concordance.
- Naively implemented, the complexity of the Star algorithm for GLMs could potentially scale exponentially with the dimension. A natural open question raised by the authors is whether it can be computed efficiently. In addition to this direction, there is another approach to improper logistic regression besides the mixture/Vovk AA and Star-shaped aggregation, see:
Mourtada, Gaiffas "An improper estimator with optimal excess risk in misspecified density estimation and logistic regression",
which is based on regularization using the test sample and is computed at the cost of finding a convex ERM.
- The statement of Proposition 1 is perhaps too vague at this stage in the text, as the setting is not properly introduced yet.
- Equation (21): the dependence on log(1/rho) should be in principle be additive rather than multiplicative.
The same should hold for Corollary 18.
- Line 261: "so we optimal"
- For square loss, the constant 1/9 is obtained; if I remember correctly, this is actually the optimal constant for this inequality (as can be seen by considering aligned points?).

--
Update:
Thank you for your reply.
The authors addressed my comments, and in light of the reply and other reviews, I would like to maintain my positive evaluation of this paper.

**Time Spent Reviewing:**

6

---

> ### Author Response · Authors · 2021-08-07
> **Response to Reviewer TjwQ**
>
> Thanks for your review of the paper and for your helpful comments. We're glad you found it interesting!
>
> ---
> ### Simulated minimax paper
> We appreciate you calling our attention to the paper of Mourtada and Gaiffas. It was a very interesting read and definitely deserves mention and comparison as a provably efficient, improper procedure attaining fast rates for logistic regression.
>
> A notable distinction between star aggregation and both the simulated minimax procedure as well as the algorithm of Foster et. al. is that the latter two require recomputation with each prediction query.
>
> In contrast, star aggregation finds a *fixed* model aggregate that can be evaluated in $O(1)$ time per input. We believe this is good motivation for the question of whether it can be estimated efficiently.
>
> ---
> ### Self-concordance in abstract
>
> We agree that without additional structure, self-concordant functions are not shown to be $(\mu, d)$-convex with respect to a fixed metric $d$, and will temper our claims accordingly.
>
> ---
> ### Proposition 1
> Following your comments as well as those of Reviewers AiEE and 2t7n,
> we plan to expand our discussion of GLMs in the intro and replace Proposition 1 (which is restated in more generality in Section 5) with an more informal description of the result.
>
> Hopefully this will be better suited to the introduction and will provide more meaningful context.
>
> ---
> ### Additive dependence on $\ln(1/\rho)$
>
> The dependence on $\ln(1/\rho)$ can be made additive in Eqn. (21), and in fact our proof recovers the stronger result; we will strengthen our claim accordingly.
>
> In Corollary 18, we are able to bound the sub-gaussianity parameter of the fluctuations in the chaining argument as $\sigma^2 = \gamma^2/n$. So we end up with an additive term $(1/n + \gamma/\sqrt{n})\ln(1/\rho)$ which is an improvement only when $\gamma = 0$ (in VC-type classes).

---

### Official Review · Reviewer_2t7n · 2021-07-18

**Rating:** 7
**Confidence:** 3

**Summary:**

This paper provides a unifying framework to study convex empirical risk minimization as well as improper learning under general entropy conditions. The key contribution is to prove a generalized version of the offset Rademacher process which takes into account a general notion of convexity, called $(\mu, d)$-convexity.

**Limitations And Societal Impact:**

Yes, the limitations have been described

**Main Review:**

Overall, the paper makes an interesting observation regarding the fast rates in supervised learning. Seemingly disparate conditions of exp-concavity and self-concordance can be viewed in a unified framework.

There are a few points/questions worth highlighting.
- One weakness of this work currently seems to be that while it is able to unify existing known upper bounds, it rarely shows applications where rates previously unknown could be derived as a consequence of this unification. One exception is the improvement mentioned in line 248 for the log-loss, but I would like to see that written out more explicitly and highlighted in the main text.
- Are there possible scenarios where this generalized offset process does not provide the right learning rates even though the underlying functions satisfy $(\mu, d)$-convexity?
- In terms of writing, I feel that the paper could substantially improve its readability by providing a notation section which summarizes the various notation used in the paper. For instance, line 144 has a notation $f_i^*, \hat{f}_i$ which is never defined (the i might also be a typo here). Example 1 uses a function $\psi$ which is not defined for a single input. Line 161 $h(Z)$ is not defined.

**Time Spent Reviewing:**

2

---

> ### Author Response · Authors · 2021-08-07
> **Response to Reviewer 2t7n**
>
> Thank you for your helpful feedback on the paper and time spent reviewing!
>
> ---
>
> ### Notation
> Firstly, we apologize for the inconsistent notation and will be sure to thoroughly correct and standardize notation. A glossary of notation is a fantastic idea that we plan to incorporate, space permitting.
>
> ---
>
> ### Log loss
> In response to your other comments, and building on the discussion with Reviewer AiEE, it seems that further expanding on the log-loss example will be very beneficial to the paper.
>
> As far as improved rates, the gist is that while the Lipschitz constant of the log loss over $[\delta, 1]$ grows as $\delta^{-1}$ as $\delta \downarrow 0$, the range only grows as $\ln(1/\delta)$. This results in an exponential improvement in the dependence on the parameter $\delta$. Performing regularization with $\delta$ chosen optimally to balance approximation and sampling error, the approximation error goes from being $\Omega(1/\sqrt n)$ in the first case to $O(\ln(n)/n)$ in the second case.
>
> Since our results hold in the improper setting, this opens the door to new rates for the star algorithm in convex and non-convex generalized linear models.
>
> ---
> ### Sub-optimality
> The log loss also gives us a good example of the sub-optimality of our results: The "price" of this improvement, at least in our analysis and treatment of exp-concave losses, is that we use the entropy of $\psi \circ \mathcal F$ instead of that of $\mathcal F$.
>
> Thus, the Lipschitz constant of the loss does show up in the entropy numbers of the class. In VC-type classes, this only costs logarithmic factors. However, in very large classes it costs polynomial factors and our results are known (by a non-constructive argument due to Bilodeau et. al., ICML 2020) to be suboptimal.
>
> ---
> We propose summarizing this aspect of our contribution, and its limitations, in more detail at the beginning of the paper. This should also provide more context for our Proposition 1.

---

### Official Review · Reviewer_AiEE · 2021-07-21

**Rating:** 6
**Confidence:** 3

**Summary:**

The paper provides a unified analysis of fast rates in statistical learning. In particular, they provide excess risk bounds for the star-ERM solution for various loss functions by bounding the offset Rademacher complexity. In particular, they identify a general property: the suboptimality gap of a point with respect to the star-ERM solution is lower bound by distance (under a particular metric) between that point and the star-ERM solution. They derive this metric for various assumptions on the loss functions like uniform convexity, exp-concavity, self-concordance, etc. In terms of new results, using these techniques:
(a) they can obtain fast rates for l_p losses for p > 2.
(b) they can obtain fast rates for exp-concave losses that depend on the range of the loss instead of its Lipschitz constant.


**Main Review:**


I think the paper provides a comprehensive summary of the machinery of offset Rademacher for obtaining fast rates, and provide new results for some settings. However, I feel that the new contributions are limited, and thus I can not support strongly for acceptance. I also feel that the paper is written in a very contrived manner and it is hard to understand what the new results are and why they are important. A few illustrative examples with a proper comparison to prior work might help in delivering the message of the paper more clearly.


**Time Spent Reviewing:**

5 hours

---

> ### Author Response · Authors · 2021-08-07
> **Response to Reviewer AiEE**
>
> Thank you for your feedback! We will work to make sure that our contributions are transparently stated and can be more easily evaluated by the reader.
>
> ---
>
> ### Novelty of contributions
> As we see it, our main contribution is methedological: we provide a clean and unified analytic tool and algorithm for obtaining fast convergence rates in a wide variety of settings: both under different conditions on the loss function (uniform convexity, exp-concavity) and on the class of functions (convex vs. non-convex).
>
> Aside from new convergence rates, we hope that this provides a useful conceptual and analytic framework for understanding the vast literature on fast rates in stochastic optimization under  entropy conditions, and for applying it to new settings.
>
> ---
>
> In addition to the results for $\ell^p$ and exp-concave losses you mentioned, we believe our concluding observations on generalized linear models are of significant interest: we show that a unified improper procedure that outputs a fixed mixture of at most three models (regularized star aggregation) enjoys a fast rate with respect to any (possibly misspecifed) generalized linear model.
>
> We plan to exhibit improper logistic regression as a self-standing example to illustrate the utility of these results.

---

### Decision · Program_Chairs · 2021-09-27

**Decision:**

Accept (Poster)

**Comment:**

The paper provides a nice unifying method to obtain fast rates for exp-concave losses, uniformly convex losses etc via an offset rademacher complexity type term. While the rates covered in the paper are not new, the unifying analysis is simple and clean.

The reviewers are positive about the contributions of the paper and think that the paper is worth publishing. I concur with the reviews and find the contributions of the paper clean and neat. Overall I recommend acceptance.